# Benchtop mesoSPIM: a next-generation open-source light-sheet microscope for cleared samples

Nikita Vladimirov [1,2,3,15] ✉, Fabian F. Voigt [1,4,13,15], Thomas Naert[5], Gabriela R. Araujo[6], Ruiyao Cai[7,8,14], Anna Maria Reuss[4,9], Shan Zhao[10], Patricia Schmid[5], Sven Hildebrand [11], Martina Schaettin[4,12], Dominik Groos [1], José María Mateos [3], Philipp Bethge [1,4], Taiyo Yamamoto [5], Valentino Aerne[6], Alard Roebroeck [11], Ali Ertürk[7,8], Adriano Aguzzi [4,9], Urs Ziegler[3], Esther Stoeckli [2,4,12], Laura Baudis[6], Soeren S. Lienkamp [5] & Fritjof Helmchen [1,2,4] ✉

In 2015, we launched the mesoSPIM initiative, an open-source project for making light-sheet microscopy of large cleared tissues more accessible. Meanwhile, the demand for imaging larger samples at higher speed and resolution has increased, requiring major improvements in the capabilities of such microscopes. Here, we introduce the next-generation mesoSPIM ("Benchtop") with a significantly increased field of view, improved resolution, higher throughput, more affordable cost, and simpler assembly compared to the original version. We develop an optical method for testing detection objectives that enables us to select objectives optimal for light-sheet imaging with large-sensor cameras. The improved mesoSPIM achieves high spatial resolution (1.5 μm laterally, 3.3 μm axially) across the entire field of view, magnification up to 20×, and supports sample sizes ranging from sub-mm up to several centimeters while being compatible with multiple clearing techniques. The microscope serves a broad range of applications in neuroscience, developmental biology, pathology, and even physics.

Tissue clearing[1] and light-sheet imaging[2] are both century-old techniques that have garnered significant attention in the past two decades[3,4]. The combination of tissue clearing with light-sheet microscopy[5–9] opened up multiple possibilities in neuroscience[10–14], developmental biology[15–17], and other biomedical fields[18–20]. These techniques allow researchers to visualize the anatomical structures of entire organs and even whole animals in three dimensions (3D) with high speed, contrast, and resolution, all without the need for labor-

[1]Brain Research Institute, University of Zurich, Zurich, Switzerland. [2]University Research Priority Program (URPP), Adaptive Brain Circuits in Development and Learning, University of Zurich, Zurich, Switzerland. [3]Center for Microscopy and Image Analysis (ZMB), University of Zurich, Zurich, Switzerland. [4]Neuroscience Center Zurich (ZNZ), University of Zurich, Zurich, Switzerland. [5]Institute of Anatomy and Zurich Kidney Center (ZKC), University of Zurich, Zurich, Switzerland. [6]Department of Physics, University of Zurich, Zurich, Switzerland. [7]Institute for Tissue Engineering and Regenerative Medicine (iTERM), Helmholtz Center Munich, Neuherberg, Germany. [8]Institute for Stroke and Dementia Research, Klinikum der Universität München, Ludwig-Maximilians University Munich, Munich, Germany. [9]Institute of Neuropathology, University Hospital Zurich, Zurich, Switzerland. [10]Department of Quantitative Biomedicine, University of Zurich, Zurich, Switzerland. [11]Department of Cognitive Neuroscience, Faculty of Psychology & Neuroscience, Maastricht University, Maastricht, The Netherlands. [12]Department of Molecular Life Sciences, University of Zurich, Zurich, Switzerland. [13]Present address: Department of Molecular and Cellular Biology, Harvard University, Cambridge, MA, USA. [14]Present address: Department of Biology, Stanford University, Stanford, CA, USA. [15]These authors contributed equally: Nikita Vladimirov, Fabian F. Voigt. ✉e-mail: vladimirov@hifo.uzh.ch; helmchen@hifo.uzh.ch

intensive tissue sectioning. The currently available techniques allow labeling and clearing of tissue samples that span multiple centimeters in size, including the whole mouse[21,22], as well as entire human organs such as eyes, kidneys, and even brains[23]. On the other hand, imaging smaller samples in large quantities increasingly finds applications in pathology workflows that are shifting from 2D to 3D[24–26]. This spectrum of applications requires both high resolution *and* large field of view (FOV)−two conflicting requirements in microscopy−to speed up the acquisition, minimize stitching artifacts, and maximize the information content of the image.

The challenge of maximizing resolution while increasing the FOV is not specific to microscopy. To maximize the FOV, astrophotography[27] and the machine vision industry currently employ CMOS cameras from 25 up to 600 megapixels and sensor diagonals from 35 up to 115 mm (e.g., Teledyne Photometrics COSMOS-66). Notably, several microscopy projects developed high-resolution, large field-of-view systems as well. The AMATERAS[28] system employed a 122 MP CMOS camera with 35 mm diagonal and a 2× telecentric lens, and an optical resolution of 2.5 μm for wide-field imaging of rare cellular events. Another example is the custom-made Mesolens[29] objective that provides a lateral resolution of 0.7 μm across a 24 mm field (sensor side), for epi-fluorescence or confocal microscopy. Most recently, the ExA-SPIM[30] light-sheet microscope achieved an extremely large FOV with a machine vision lens and a 151 MP CMOS camera (sensor diagonal 66 mm), with lateral and axial resolution of 1 and 2.5 μm, respectively.

The trend toward larger sensors in microscopy continues, and several large-sensor sCMOS cameras are now available with programmable rolling shutter ("light-sheet") readout mode that are suitable for mesoSPIM imaging (diagonals of 25–29 mm, e.g., Teledyne Photometrics Iris 15 and Kinetix, Hamamatsu Orca Lightning). However, these cameras are already incompatible with most life science microscope objectives, which are designed for smaller fields of view (18–25 mm), resulting in unsatisfactory image quality at the periphery. This issue can be resolved by using improved optics that meet the standards set by modern cameras. The scientific community therefore is actively developing methods for comparing and reporting properties of microscopes and their components which are relevant to end users[31,32]. Currently, methods to quantify the performance of detection optics in a light-sheet microscope independently from the properties of the light-sheet excitation are rarely employed. As a result, the optimal selection criteria for a detection objective suitable for large sensors are unclear.

The mesoSPIM project[33] provides instructions for building and using a facility-grade light-sheet microscope for imaging large, cleared samples in a free and open-source way. The mesoSPIM system achieves uniform axial resolution across cm-scale FOV by using the axially swept light-sheet microscopy principle (ASLM)[33–36]. In brief, the axially most confined region of the light-sheet (the excitation beam waist) is moved through the sample in synchrony with the camera's programmable rolling shutter by using an electrically tunable lens (ETL) as a remote focusing device. The synchrony of the beam translation and the camera readout leads to uniform axial resolution across the FOV (see reviews in refs. 36,37). Our previous mesoSPIM design (v.5) has several limitations. Its detection path relies on a macro zoom microscope (Olympus MVX-10), which not only limits the system's magnification range but also makes it incompatible with the latest generation of large-sensor cameras, due to vignetting at the periphery of the sensor. The footprint of mesoSPIM v.5 is relatively large, with multiple custom parts, and requires an optical air table, thus making the microscope relatively expensive, complex to build and align for non-experts, and difficult to move to new places.

Here, we present an improved version of the mesoSPIM ("Benchtop") that features a large-sensor sCMOS camera and an optimized detection system, resulting in higher resolution across a much larger FOV. We achieved this by developing a method to quantify the optical properties of detection objectives independently of the light-sheet excitation. The improved design is more compact, more affordable, easier to build for non-experts, and it is travel-friendly. We demonstrate examples of objective performance quantification, as well as several applications of the Benchtop mesoSPIM in neuroscience, developmental biology, and even physics, where we image particle tracks in transparent crystals that work as particle detectors. We freely share full details of Benchtop mesoSPIM assembly and operation aimed at non-experts who want to build their own system.

## Results
### Overview
We are introducing the Benchtop mesoSPIM (Fig. 1a–c), which offers several advantages over its predecessor, the mesoSPIM v.5, including higher resolution, a larger FOV, and a wider range of magnifications. In addition, the Benchtop mesoSPIM significantly reduces the instrument's footprint and cost (Table 1), does not require an optical table, and can be used on a lab bench. The instrument is compatible with modern large-sensor sCMOS cameras, such as the Teledyne Photometric Iris 15, and offers a 1.9× larger sensor area and 3.6× higher pixel count per image than its predecessor (Fig. 1d). It achieves optical resolution down to 1.5 μm laterally and 3.3 μm axially, and high field flatness across the entire field (Fig. 1e, Supplementary Figs. 1 and 2). These features allow imaging of 1 cm³ sample volume in as little as 13 min (see Supplementary Note 1), making it possible to image large biological samples at high speed.

To increase the system throughput and simplify sample mounting, we designed a range of sample and cuvette holders that can be 3D printed from chemically resistant plastics (polyamide PA12, see Fig. 1f). These holders are compatible with various clearing and immersion chemicals such as benzyl alcohol/benzyl benzoate (BABB), dibenzyl ether (DBE), and ethyl cinnamate (ECi). They are also easy to clean and autoclave. The holders can accommodate samples ranging from 3 to 75 mm in length. There are four classes of holders available (from left to right in Fig. 1f): clamps for direct sample dipping, holders for glass cuvettes that separate the immersion and imaging media (including extra-safe holders for BSL-3 biosafety level), high-throughput multi-sample holders (the "SPIM-tower"), and holders for non-dipped large cuvettes for extremely large samples, such as whole mice. With this diverse set of holders, our system provides high versatility for imaging a variety of samples.

Imaging the distribution of fluorescently labeled neurons in the mouse brain and their projections can be done with the Benchtop mesoSPIM at single-axon resolution when using sparsely labeled neuronal populations. An example image of a Thy1-GFP line M[38] mouse brain with spinal cord, cleared with vDISCO[21], shows that axons and dendrites are detectable across the brain at 5× magnification, as shown in Fig. 2.

Like its predecessor, the Benchtop mesoSPIM remains compatible with a wide range of clearing methods due to the use of low-NA long-working distance (WD) air objectives. It is capable of imaging large clear samples due to its long-range translation stages with 50 × 50 × 100 mm travel, covering up to 250 cm³ volume.

Based on the configurations outlined here, users can choose a combination of detection objective, tube lens, and camera that provides a system resolution tailored to their needs. An informed choice of these fixed optical components is thus critical for good system performance.

**Testing of microscope objective contrast properties**
Quality control of microscope objectives by the manufacturers is typically done manually using interferometric methods, which requires special equipment and skills. The microscopy community developed its own testing methods that aim to estimate resolution, field flatness, and other parameters across the imaging field by

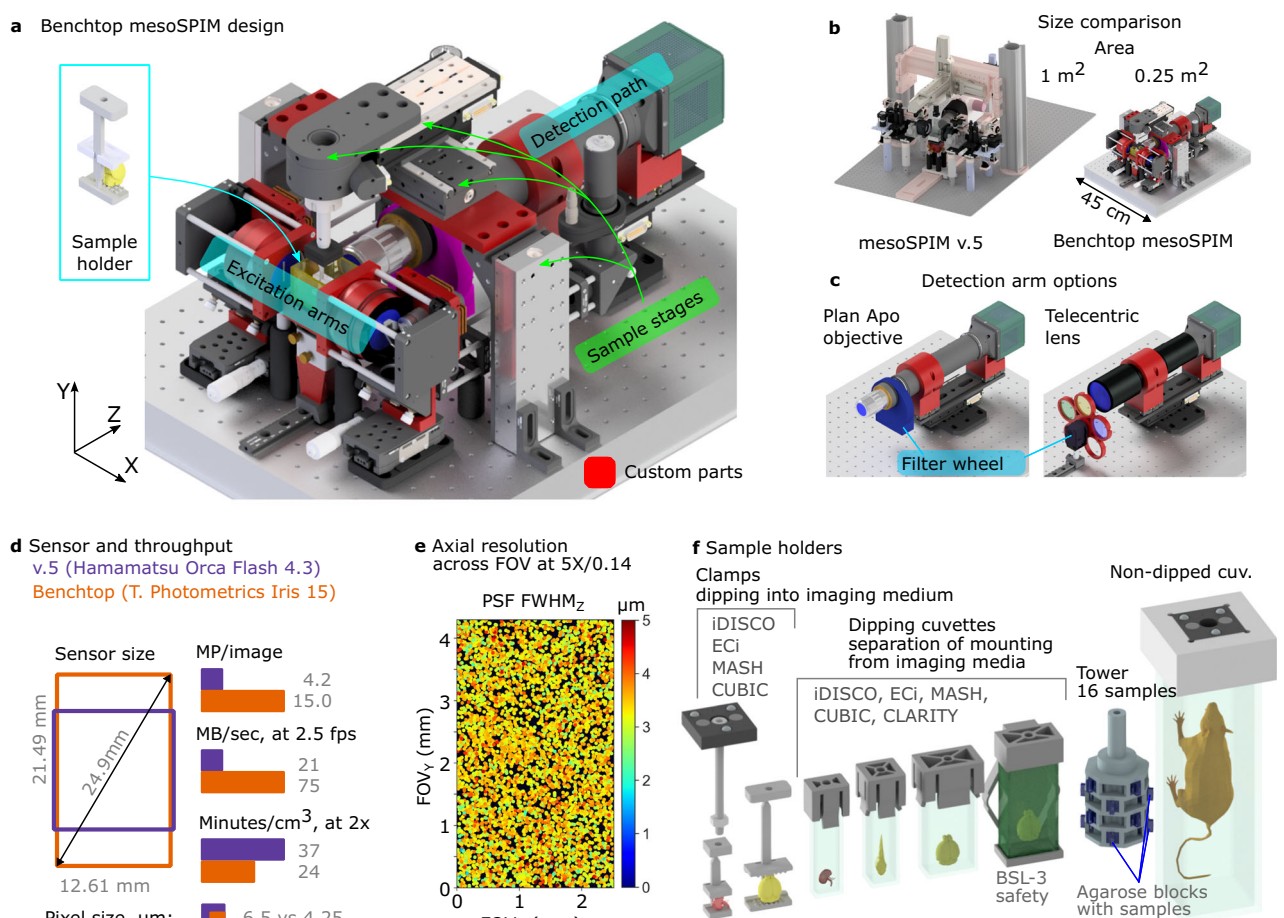

**Fig. 1 | The Benchtop mesoSPIM design and an application example. a** CAD model of the microscope with main modules labeled: excitation arms, detection path, sample stages, and sample holder. Modified or custom-made parts are red. **b** Size comparison between mesoSPIM v.5 and Benchtop systems. **c** Detection arm can be equipped with a plan apochromatic objective (2×–20× magnification, tube lens not shown) or a telecentric lens (0.9×–2×) depending on the application, with a corresponding filter wheel and set of filters. The detection arm is mounted on a focusing stage. **d** Comparison of sensor size, pixel size, pixel count per image, and imaging throughput between v.5 (Hamamatsu Orca Flash4.3 camera) and Benchtop (Teledyne Photometrics Iris 15). **e** The Benchtop mesoSPIM axial resolution in ASLM mode across the FOV, at magnification 5× and NA 0.14. The full width at half-maximum of the point-spread function along the z-axis (FWHM$_Z$) is color-coded from 0 to 5 μm. The resolution was measured with 0.2 μm fluorescent beads embedded in agarose immersed high-index medium (RI = 1.52). **f** CAD models of custom 3D printed sample holders that accommodate samples from 3 mm to 75 mm.

mounting fluorescent beads on slides (or embedding them in agarose) or by using commercially available calibration structures[32,39].

In light-sheet microscopy, imaging of fluorescent beads embedded in agarose and index-matched with the immersion medium and measuring the point-spread function (PSF) size is currently the gold standard[36]. However, the results of such a test depend on both excitation and detection optics since the properties of the illumination beam are critical for the system's axial resolution. In light-sheet microscopes that use ASLM scanning, such as mesoSPIM and ctASLM[35], resolution measurements are also affected by the synchronization of the rolling shutter with the axial motion of the beam, by the exposure time, and by the lateral scanning frequency of the beam[36]. As a result, it is difficult to separate the detection objective properties from excitation parameters by measuring the system PSF from images of fluorescent beads.

To address this problem, we drew inspiration from modern methods of full-field camera lens testing[40,41] to evaluate the modulation transfer function (MTF) across the full camera field and full focus range, resulting in 3D MTF graphs measured at specific spatial frequencies (e.g., 40 lp/mm, line pairs per mm) and angular orientations. To test microscope objectives, we adapted this method by using a high-density Ronchi grating slide at a horizontal line orientation. The

slide was uniformly illuminated with an incoherent light source (a smartphone screen with an even white field), and a focus stack was captured with a 10-μm step size between planes.

The air objectives were tested in two conditions: first, with the Ronchi grating in air, and then with the Ronchi grating immersed in oil with a refractive index of 1.52, ~20 mm thick between the objective and the sample (as shown in Fig. 3a). This oil immersion mimics the conditions of imaging cleared tissue in a mesoSPIM setting as it introduces spherical aberration from the immersion medium and cleared tissue that is typically present in the system. The described testing method provides quantitative 3D contrast maps along the x, y, and focus dimensions of a detection objective (Fig. 3b and Supplementary Movie 1), allowing for the computation of compound performance metrics, such as field flatness (best-focus surface sag), depth of field, maximum contrast, and contrast variation across the field. This approach is low-cost, applicable to all types of objectives and immersion media, independent of light-sheet illumination, and can be completed in just a few minutes per objective.

To our surprise, the contrast maps of the Olympus MVPLAPO-1× objective with zoom body MVX-10 (the default mesoSPIM v.5 configuration) showed poor centricity and large variation of contrast across the field at some zoom settings, especially at 1.25× and 2× zoom

**Table 1 | mesoSPIM v.5 vs Benchtop comparison**

| | Version 5 | Benchtop |
|---|---|---|
| **Optical resolution, axial** | 5.0 µm | 3.3 µm |
| **Optical resolution, lateral** | 2.7 µm[r1] | 2.0 µm (1.5 – 2.6 µm)[r2] |
| **Magnification range** | 0.63× – 6.3× | 0.9× – 20× |
| **Detection objectives** | Variable (motorized zoom) | Fixed-magnification (manual exchange) |
| **Camera sensor diagonal (equiv. to FOV at 1× magnification)** | 19 mm | 25 mm |
| **Pixel size** | 6.5 µm | 4.25 µm |
| **Pixels/image** | 4 MP | 15 MP |
| **Footprint** | 1 m² | 0.25 m² |
| **Mobile** | no | yes |
| **Approximate parts cost, USD** | 162,000[c1] | 100,000[c2] |

r1, resolution estimated with Olympus MVPLAPO 1× objective at 4× zoom[33].

r2, resolution averaged across three Mitutoyo BD Plan objectives: 5×: 2.6 µm, 10×, 1.8 µm, 20×, 1.5 µm, see Supplementary Fig. 1 for details.

c1, Version 5 costs include 4 laser lines (405, 488, 561, 638 nm) and optical table.

c2, Benchtop cost estimate includes 3 laser lines (488, 561, 638 nm). Fourth laser line (e.g., 405 nm) is optional, at an additional cost of about 6000 USD. Magnifications included in the price: 2×, 5×, 10×. The cost estimate does not include PC workstation and a screen. Costs are subject to change by vendors.

(Supplementary Fig. 3), presumably because of the aberrations induced by the immersion medium (see below). We thus searched for fixed-magnification objectives as an alternative, which are known to provide higher image quality at lower cost, due to the absence of moving elements and thus higher alignment precision of the lens groups. To this end, we tested 11 long-WD plan apochromat air objectives from Olympus, Thorlabs, Mitutoyo, and other companies, and found their contrast performance typically degraded to various degrees at the field periphery (Supplementary Figs. 2–4 and Supplementary Table 1).

We found that industrial objectives from Mitutoyo (specifically the Plan Apo BD, M, and G series) provided the best contrast and field flatness. These objectives demonstrated excellent performance in terms of field flatness, contrast uniformity, and absolute contrast values (as shown in Supplementary Fig. 2), while being more affordable than life science microscopy objectives. When compared to the Olympus MVPLAPO-1× at the same effective magnification, the contrast maps of the Mitutoyo objectives were significantly better in terms of their absolute values, uniformity, centricity, and the resulting image quality of a biological sample (as shown in Figs. 3c and 4). Furthermore, the availability of Mitutoyo long-WD air parfocal objectives at 2×, 5×, 7.5×, 10×, and 20× magnifications makes this line optimal for a wide range of mesoSPIM applications. The Mitutoyo Plan Apo BD and M series are equivalent in their magnification, working distance, and NA.

Other objectives which scored high in our tests were from the Thorlabs super apochromat series and the Olympus XLFluor 4×/0.28. For applications that require lower magnification (e.g., for fast screening of mouse brains with single-cell resolution and no tiling), we tested and used industrial telecentric lenses with 0.9× and 1.2× magnification (Supplementary Table 1).

The testing method described above uses white light for Rochi slide illumination, where the smartphone screen emits red, green, and blue LEDs (emission spectrum shown in Supplementary Fig. 5a). This restriction limits the quantification of chromatic effects that would be representative of fluorescence channels. To quantify the role of chromatic effects, we therefore expanded our testing method with a tungsten-halogen lamp and a set of 6 bandpass filters (spectra shown in Supplementary Fig. 5b), and a Ronchi slide immersed in DBE (20 mm medium before the slide).

As a control, we imaged the Ronchi slide in the air using the Olympus MVPLAPO-1× objective at 2× zoom of MVX-10 body and found a relatively flat field at all chromatic channels (Supplementary Fig. 6a), indicating that the objective performed nominally. This was in stark contrast with our earlier measurements when the Ronchi slide was immersed in oil and illuminated with white light (Supplementary Fig. 3). Indeed, we could reproduce the strongly curved field again when the slide was immersed in DBE (Supplementary Fig. 6b), which indicates that the best-focus surface becomes curved in the presence of immersion medium. This presumably occurs because the objective is not telecentric at low zoom, so the combination of spherical aberration, coma, and astigmatism distort the best-focus field and render it non-flat. Additionally, large chromatic focal shifts are visible with the DBE-immersed Ronchi slide. Notably, focus offsets relative to the shortest wavelength (420 nm) were positive in DBE but negative in air, presumably due to DBE chromatic dispersion.

We found that under our emulated "cleared tissue" conditions, both the Olympus MVPLAPO-1× objective and the Mitutoyo objectives have significant chromatic focal offsets (i.e., best-focus planes vary for different channels), up to 400 µm between blue (420/20 filter) and red (697/75 filter) channels (Supplementary Fig. 6b–e). For example, the focal plane offset between GFP (535/22 filter) and RFP (630/69 filter) channels can be between 50 and 150 µm, positive or negative, depending on the objective. In most cases, the dependence of the field flatness on the chromatic channel was rather weak (except MVPLAPO-1× at zoom 2×), which suggests that channel-dependent focus offset is the main factor to consider for optimal imaging performance.

In practical terms, the mesoSPIM detection objective must be focused differently for each channel, which is achieved through the control software (*Focus* button group, Supplementary Fig. 8). The amount of refocus depends on the channel, medium, and objective and is adjusted manually for each specific set of conditions.

## Control software

The control software is an open-source Python program[42] that is user-friendly and compatible with all versions of mesoSPIM (Supplementary Fig. 8). It supports various types of hardware, such as cameras, stages, and filter wheels (Supplementary Tables 4–6). The underlying PyQt5 platform allows for multi-threading to efficiently handle imaging data, while specialized libraries like npy2bdv[43] enable fast saving of multi-tile/channel/illumination acquisitions in Fiji[44] BigDataViewer[45] HDF5/XML and other file formats, streamlining stitching, fusion, and visualization. This format enables terabyte-sized datasets to be acquired and processed on a local workstation with modest RAM requirements (128 GB). The software is modular and allows for system upgrades by modifying a single configuration file.

## Examples of application

Imaging of a Thy1-GFP line M[38] mouse brain with spinal cord demonstrates that individual axons and dendrites are resolved already at 5× magnification, as shown in Fig. 2 and Supplementary Movie 2 (acquisition details in Supplementary Table 3). When higher magnification is required, the Benchtop system can be equipped with a 20× air objective. Figure 5a shows two pyramidal neurons in a mouse prefrontal cortex, with basal dendrites and axons clearly visible. Notably, the detection objective (Mitutoyo Plan Apo G 20×/0.28(t3.5)) is precompensated by design to image through 3.5 mm of glass-like medium (RI 1.52), which improves image quality by reducing spherical aberration caused by the medium mismatch.

The currently available clearing protocols enable labeling and imaging of cells and their projections throughout the entire body of the mouse[21,22]. Our system is capable of imaging a whole mouse (P14 age, at 0.9× magnification, Fig. 5b and Supplementary Movie 3), potentially allowing for the creation of a whole-mouse digital atlas of cell types[22] or tracking the spread of metastases[21].

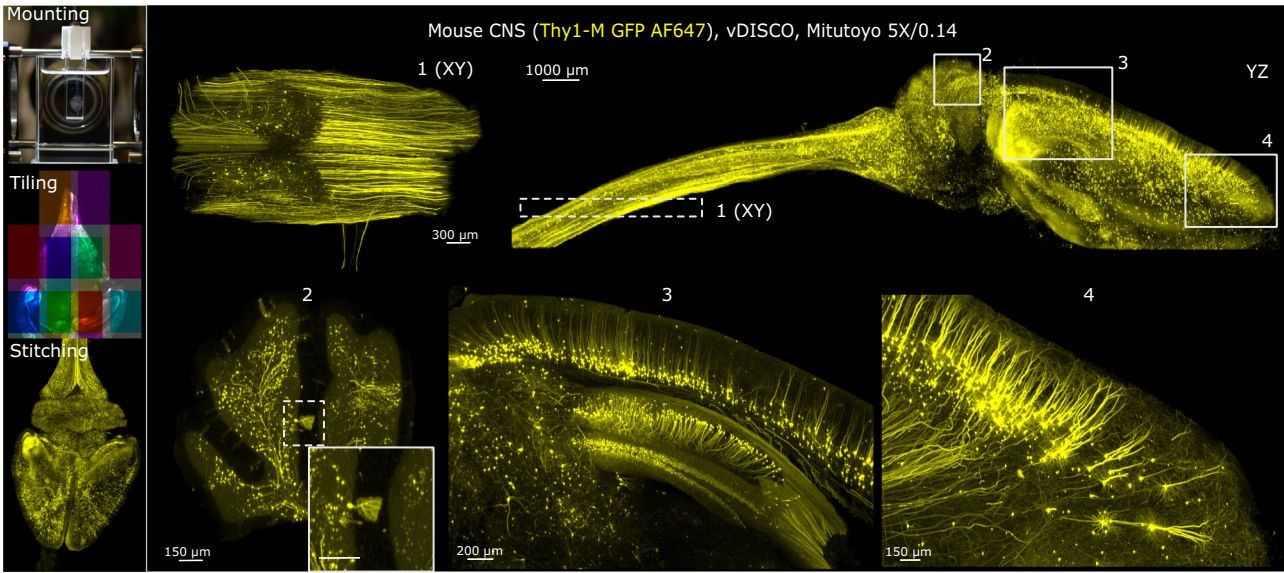

**Fig. 2 | Example of a multi-tile imaging of a mouse CNS at 5× magnification.** The workflow consists of sample mounting, tiled imaging, dataset stitching and fusion. The brain and spinal cord of a mouse Thy1-GFP line M, stained with Atto647N conjugated anti-GFP nanobooster and cleared with vDISCO, was imaged with a 5×/0.14 detection objective. The data shows axonal and dendritic arborizations of long-projecting neurons: motor axons in the spinal cord (inset **1**), Purkinje cells (inset **2**), pyramidal cells in the cortex and hippocampus (inset **3**), pyramidal cells in the prefrontal cortex (inset **4**). Maximum intensity projections (MIPs) over multiple planes spanning 500 μm (insets **1**, **2**), 1000 μm (insets **3**, **4**), or the entire volume (YZ projection of the brain) are shown. Gamma correction (1.5 in Imaris) was applied. The original grayscale 16-bit map was pseudo-colored with a yellow look-up table (LUT). The imaging experiment was performed twice with nearly identical results.

Development of the vertebrate nervous system, and especially axonal growth, is often studied using chicken embryos as a model[46]. As an example of imaging the nervous system *in toto*, a neurofilament-stained chicken embryo at E9 cleared with BABB[5] is shown in Fig. 5c, Supplementary Movie 4.

The regional and temporal heterogeneity of amyloid-β (Aβ) plaque formation in mouse brains subject to various treatments is a promising direction in Alzheimer's disease research[13]. An APP/PS1 mouse brain cleared with iDISCO[47] is shown in Fig. 5d, Supplementary Movie 5) at a magnification of 1.2×. Amyloid plaques are displayed in green, blood vessels in magenta.

The neuronal projections to specific areas of the mouse brain can be labeled by injecting a retrograde AAV virus inducing fluorescent protein expression. As an example, we injected a Cre-dependent td-Tomato expressing viral construct into the lateral habenula (LHb) of a Vglut2-Cre mouse and imaged the axonal and dendritic arborizations of pyramidal cells projecting to this region (Fig. 5e, Supplementary Movie 6). This gives sparser and more specific neuronal labeling than, for example, Thy1-YFP expression (the latter is shown in Supplementary Movie 7 for mouse hippocampus).

In the human brain, areas V1 and V2 of visual cortex the most thoroughly studied regions and are the only two brain areas in neocortex with a border between them macroscopically visible in histological sections, which is defined by the end of the stria of Gennari. This anatomical landmark is also visible in other imaging modalities, such as magnetic resonance imaging (MRI), making the V1/V2 border a promising target for validating high-resolution post-mortem ultra-high field (UHF) MRI data using light-sheet microscopy. However, despite V1 and V2 being highly studied areas, reliable estimates of cell densities per mm³ per layer are still missing, with published estimates varying widely. High-throughput imaging of Nissl-stained samples could help to derive more reliable cell density estimates for the human brain in the future. In Fig. 5f, we show human cortex samples (occipital lobe samples excised around the V1/V2 border) from a 90-year-old male donor (Supplementary Movie 8). We also provide a dataset from a 101-year-old female donor (see Supplementary Movie 9). The tissues were imaged at 5×

magnification, cleared, and stained with either MASH-NR or MASH-MB protocol[48] (see Supplementary Figs. 9, 10 for details).

At stage 58, *Xenopus* tadpoles enter metamorphosis, a process of drastic morphological transformations and complex physiological changes. *In toto* imaging of whole animals during this transition provides a unique opportunity to uncover the molecular mechanisms governing morphogenetic changes, such as limb development, the transition from pro- to mesonephros (the adult kidney), or tail regression. Furthermore, imaging of entire intact froglets offers a comprehensive view of organs and tissues in the context that provides insights into complex interactions relevant to normal developmental processes and disease models. Supplementary Movie 10 shows a *Xenopus* tadpole at stage 58 cleared with BABB, stained with Atp1a1 antibody that labels the nervous system projections (including the olfactory system), eyes, and the developing hindlimbs of the animal at this critical developmental stage.

### High-throughput imaging of biological samples

For higher throughput imaging, we designed a modular "SPIM-tower" holder that accepts 4 samples per level, with multiple levels stacked upon each other and connected by magnets. The samples are embedded in agarose blocks that stick out of the tower frame for obstruction-free light-sheet illumination (Fig, 5g Supplementary Fig. 11). The tower is rotated in 90-degree steps and moved vertically to bring the individual samples into position for mesoSPIM imaging. The system includes a 3D printable agarose mold that allows accurate mounting of multiple samples in identical agarose blocks that are then inserted into tower slots.

The SPIM-tower enabled us to image 16 tadpoles (4 levels stacked), embedded in individual agarose blocks and cleared with BABB, in one 20-min session (Fig. 5g–j). Standardized automated whole-embryo imaging after experimental intervention, such as chemical treatment or genetic manipulation (e.g., CRISPR/Cas9), has the potential to facilitate screening efforts by distinguishing between intrinsic variability and genuine phenotypes, for example when studying effects of transient inhibition of retinoic acid (RA) signaling during embryonic development. *In toto* imaging of embryos indicates

**a**  Detection objective testing method

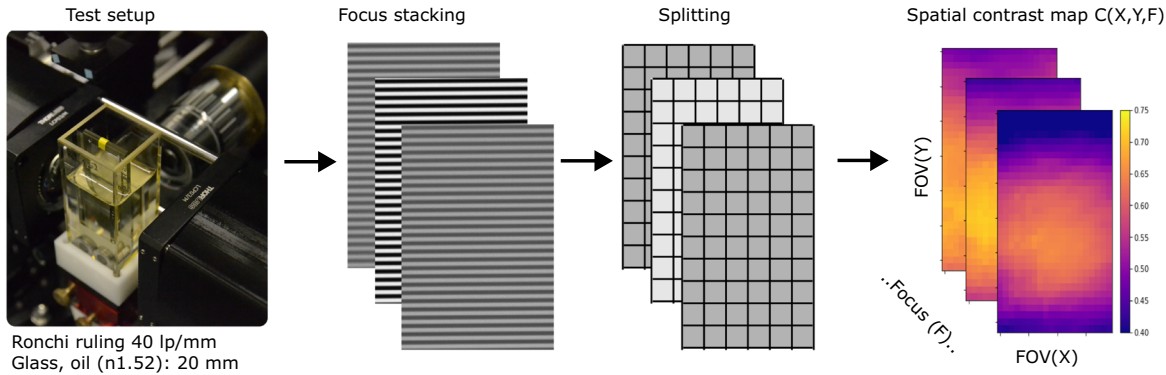

**b**  Example of a spatial contrast map: Olympus XLFluor 4x/0.28 (20 mm RI 1.52 medium)

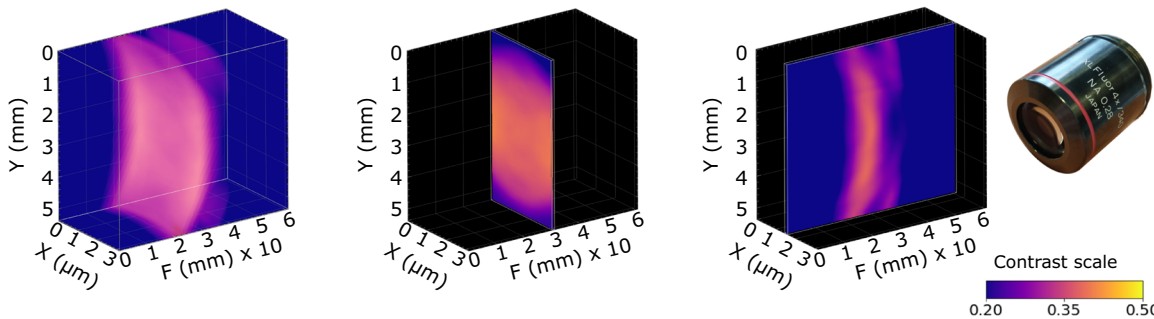

**c**  Contrast maps comparison between mesoSPIM versions
medium: 20 mm (RI 1.52)

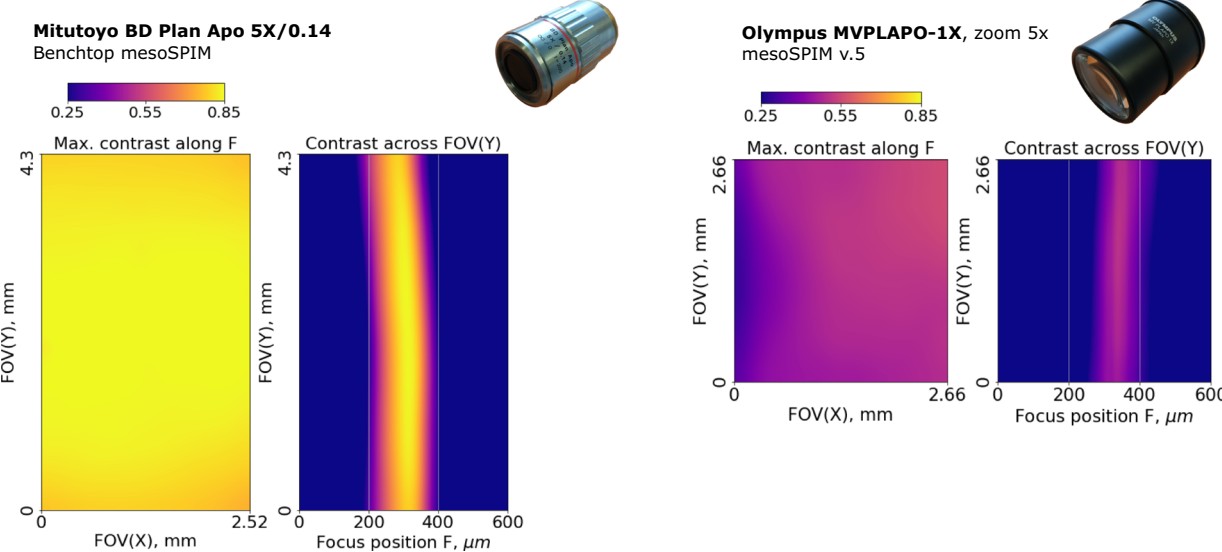

**Fig. 3 | The testing method for detection objectives. a** The testing procedure of the detection objective contrast properties includes taking a focus stack of the Ronchi ruling immersed into the imaging medium, splitting each image into sub-regions, and calculating the local contrast of each sub-region. **b** Example of a 3D contrast map for objective Olympus XLFluor 4×/0.28. **c** Comparison of absolute contrast and field flatness across the field between Benchtop mesoSPIM (Mitutoyo Plan Apo BD 5×/0.14) vs. mesoSPIM v5 (Olympus MVPLAPO-1×, at zoom 5×). Sensor dimensions and colormap scales are equal.

hypoplastic disorganized pronephros with shortened tubular lengths and retina/lens hypoplasia with effects on retinal layer organization (Fig. 5k, l). Additionally, the larger sample size of such stereotyped datasets makes them ideally suited for future deep learning-based automated phenotyping initiatives[17]. This system can be used for high-throughput imaging of embryos, organoids, and larger samples of various species.

## Imaging of color centers for particle detectors

Besides imaging biomedical samples, the high-throughput and optical sectioning capabilities of the mesoSPIM can have applications in physics and geology to study fluorescent signals from defects in transparent crystals. These fluorescence-emitting defects, so-called color centers, can be induced in crystals by irradiation[49–53]. Particles such as neutrons, ions, cosmic and gamma rays can either dislocate

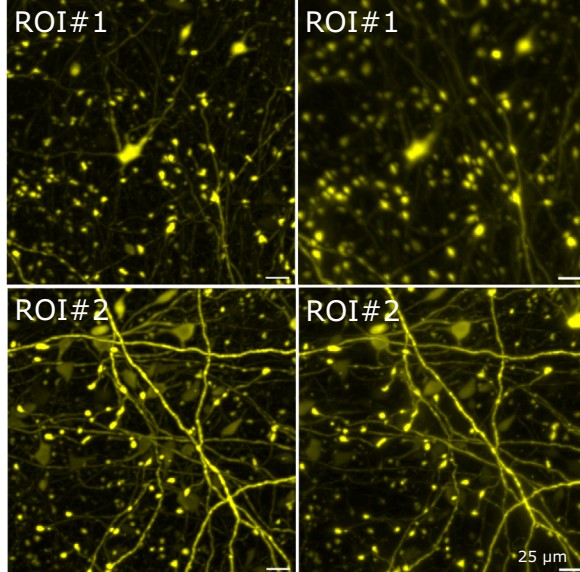

**Fig. 4 | Comparison of the FOV and image quality between Benchtop mesoSPIM and mesoSPIM v.5 at 5× magnification.** The Benchtop allows larger overall FOV and higher contrast across the image. The mesoSPIM v5 system has areas of lower resolution (ROI#**1**) and higher resolution (ROI#**2**) depending on the region's position, consistent with the maps of objective contrast shown in Fig. 3c. MIPs over multiple z-planes are shown. The original grayscale 16-bit map was pseudo-colored with a yellow LUT. Each imaging experiment was performed twice with nearly identical results.

atoms and create vacancy defects in crystalline structures and/or provide electrons to existing defects[49–53]. The resulting system can fluoresce when excited by light. By imaging these small defects with light-sheet microscopy, it is possible to probe not only the well-known interaction of particles, such as neutrons with solid-state materials but also the flux of weakly interacting particles, such as neutrinos from nuclear reactors or dark matter candidates[54–56]. Measuring the interaction of these particles requires scanning large sample volumes[54] (kilograms of transparent crystals). To study color center formation, we irradiated a 1 cm³ $CaF_2$ crystal (Fig. 6a) with gamma rays from a $^{60}$Co source which made the crystal fluorescent (Fig. 6b). When the crystal was imaged at 20× magnification, several track candidates were found repeatedly in more than one scan (which minimizes the likelihood that they are noise artifacts). One such track candidate is shown in Fig. 6c, with statistical analysis detailed in the Statistics and Reproducibility section and further in Supplementary Note 2 and Supplementary Fig. 12. While these findings are in their very early stages, the ability of the Benchtop mesoSPIM to image hundreds of cm³ of bulk crystals at high speed and resolution is instrumental for the detection of color centers induced by rare events in passive crystal detectors. These fluorescent structures can be produced by ions, neutrons, or elusive particles, such as neutrinos and dark matter candidates[54,55].

## Travel capability

The Benchtop mesoSPIM is made travel-friendly by grouping the elements into mechanically independent modules (2 excitation paths, detection path, and sample stages), which minimizes the packing and alignment requirements to ship and reassemble the system. The packed components have a volume of about 0.2 m³ and weigh 60 kg. Unpacking and reassembling the system takes less than 1 h (Supplementary Movie 11), followed by up to 1 h of test and alignment routines.

The travel capability of Benchtop mesoSPIM enables easier access to the instrument and wider collaboration opportunities between biological labs. In the context of physics (imaging rare track events), the microscope can be more conveniently used in places of difficult access, such as in underground laboratories. The laboratories are often located within mines, where the non-specific irradiation (such as cosmic rays) is minimized. For the application of nuclear reactor monitoring (discussed in Supplementary Note 2), the Benchtop mesoSPIM could be transported for in-situ sample analysis, eliminating the requirement for a dedicated setup at each location.

## Cost reduction

The substantial cost reduction of the Benchtop mesoSPIM compared to its predecessor (100k vs 165k USD) was achieved by optimizing the costs of lasers, stages, control electronics, and detection objectives. The Benchtop mesoSPIM uses a single laser combiner with up to 4 laser channels and a built-in fiber-switching module, which allows fast and reliable switching of the laser source between the two excitation arms with minimal power loss.

The compact footprint of the Benchtop mesoSPIM is achieved through several modifications. First, the macro-zoom system Olympus MVX-10 is replaced with fixed-magnification objectives. Additionally, the excitation path is redesigned to optimize space utilization. The use of more compact translation stages allows for the efficient positioning of components within a limited area. Furthermore, a low-cost compact filter wheel obtained from a hobby astronomy shop is chosen as a compact alternative. These modifications collectively contribute to the reduced size and enhanced portability of the system.

## Upgrading previous versions

The older versions of mesoSPIM (v.5 and earlier) can be upgraded in their detection path to achieve the resolution and magnification range of the Benchtop mesoSPIM. The upgrade includes a large-sensor camera, Mitutoyo objectives (2×, 5×, 7.5×, 10×, 20×), a motorized compact filter wheel, and an objective turret (Supplementary Fig. 13, Supplementary Table 7). The total cost of the upgrade including the camera is about 27k USD. All components of the excitation paths and sample stages can be used as before, thus making the upgrade cost-effective.

## Instructions for building and operation

Building a Benchtop mesoSPIM does not require elaborate skills in optics, mechanics, or programming. Customization of several metal components requires a basic metal workshop with a milling machine to modify several off-the-shelf parts. Alternatively, metal parts can be ordered online from vendors using the design files posted on the project's website. Components for which strength and long-term dimensional stability are not critical can be printed using a hobby-

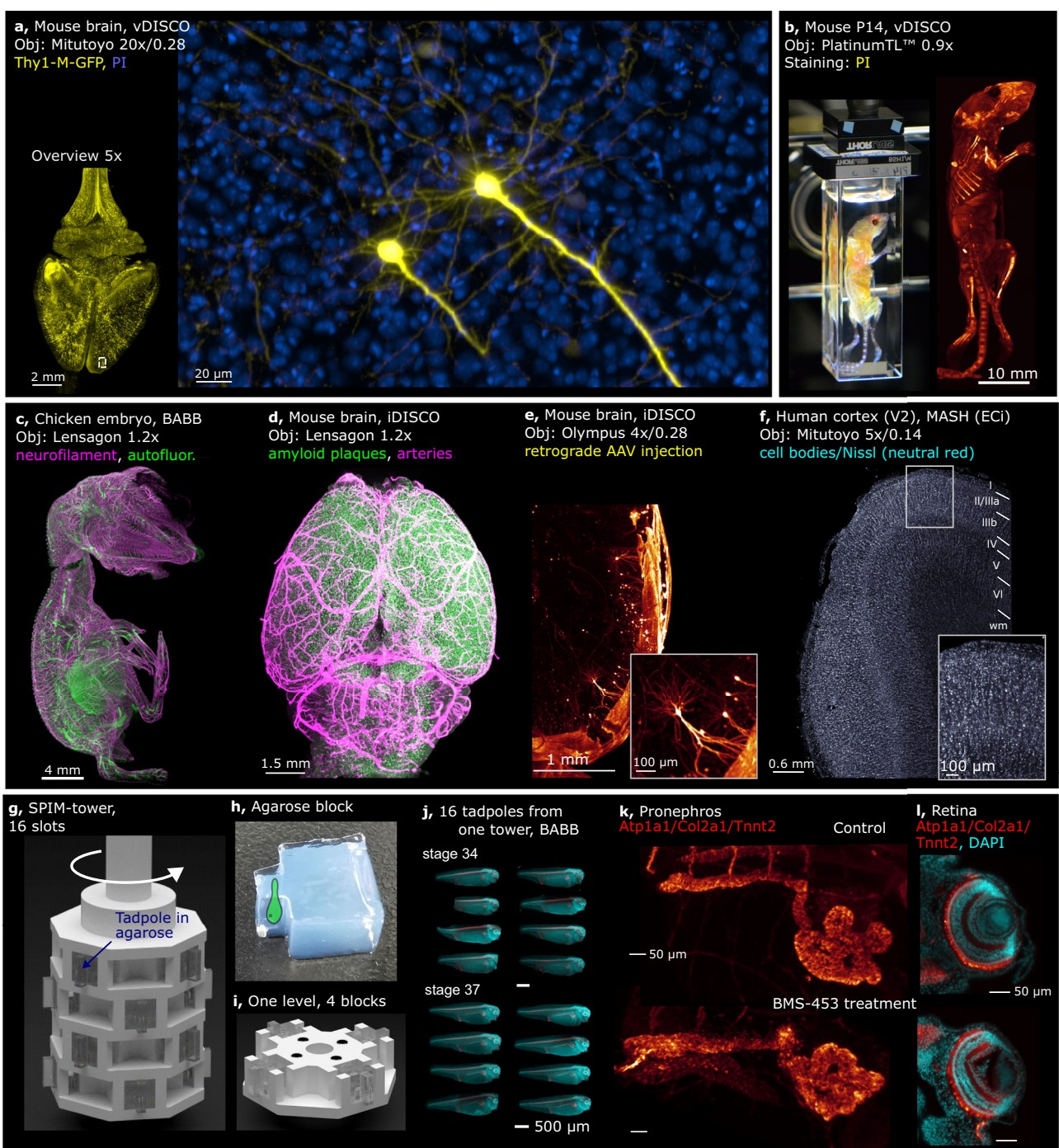

**Fig. 5 | Examples of Benchtop mesoSPIM biological applications. a** Two pyramidal cells in the prefrontal cortex imaged at 20× magnification (Thy1-GFP line M, Atto647N - yellow, propidium iodide - blue, cleared with vDISCO), with axons and basal dendrites resolved. **b** Whole mouse body imaging at 0.9× magnification (P14 mouse, stained with propidium iodide PI, cleared with vDISCO). **c** Peripheral nervous system of a chicken embryo at E9 imaged at 1.2× (neurofilament staining with mouse anti-RMO270, goat anti-Mouse Cy3; autofluorescence, cleared with BABB). **d** Mouse brain at 1.2× (APP/PS-1 line, amyloid plaques, arterial vessels, cleared with iDISCO). **e** Mouse brain at 4× (Vglut2-Cre line, sparse retrograde AAV injection for Cre-dependent td-Tomato expression, iDISCO). **f** Human brain tissue at 5× (area V2, stained with neutral red, cleared with MASH). **g** CAD model of the *SPIM-tower* sample holder. **h** A molded agarose block depicting the sample as a cartoon drawing. **i** CAD model of a single layer of the *SPIM-tower* sample holder. **j** Standardized imaging of 16 *X. tropicalis* tadpoles (stages 34 and 37) treated with BMS-453 (left) or control (right). Samples were stained with DAPI (cyan) and for Atp1a1/Col2a1/Tnnt2 (red), embedded in individual agarose blocks using the *SPIM-mold*. **k** Pronephros at stage 42 (top: control, bottom: treated), **l** Retina (top: control, bottom: treated). BMS–453 treatment appears to affect both kidney and retinal development. In all panels, MIPs over multiple planes are shown. The original grayscale 16-bit maps were pseudo-colored. Imaging experiments for cleared biological samples were performed at least once.

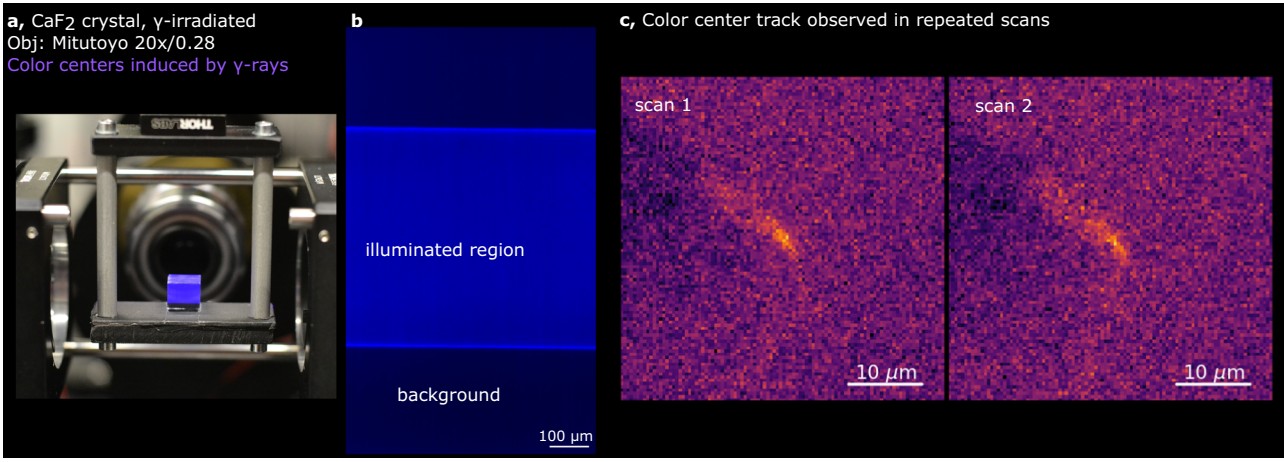

**Fig. 6 | Application of benchtop mesoSPIM for imaging of color centers.**
**a** Irradiated CaF$_2$ crystal imaged at 20× magnification (color centers induced by gamma irradiation at 5 MRad, polished crystal, no clearing); **b** raw image (color-coded blue) showing SPIM-illuminated region vs background fluorescence in the irradiated CaF$_2$ crystal. **c** Candidate track of color centers observed in repeated scans of the irradiated CaF$_2$ crystal. The structure appears across five planes (z-step 3 μm), so the maximum intensity stack is shown for each scan. The original grayscale 16-bit maps were pseudo-colored. Imaging experiments for the CaF$_2$ crystal were performed at least twice to account for noise.

grade 3D printer. Several electronic parts (e.g., electro-tunable lens and galvo drivers) require modification that involves basic soldering skills.

We offer a full parts list, purchase recommendations, wiki documentation, tutorials, and videos, for experts and non-experts alike, on how to build, align, and operate the Benchtop mesoSPIM.

## Current limitations

Due to our design choices, we use off-the-shelve air objectives in both excitation and detection arms for compatibility with multiple clearing protocols and quick exchange of imaging media, while keeping the system's cost low. Thus, the optical resolution of our current design suffers from spherical aberration from the medium index mismatch. Our simulations show that both excitation and detection arms operate in a diffraction-limited regime only with air objectives of NA up to 0.15 and immersion medium thickness up to 15 mm (Supplementary Notes 3, 4). Therefore, increasing the NA of detection objectives above 0.15 achieves higher resolution only for samples (incl. immersion medium) thinner than 15 mm (a "15–15" rule). The simulated PSF size is nearly independent of the refractive index of the medium in the practically relevant range ($n = 1.33$–1.99, Supplementary Table 11).

It should be noted that "field flatness" derived from the contrast maps of objectives in our testing method should not be confused with Petzval field curvature. The field flatness profile (2D) is computed from the best-focus surface (in 3D) of the objective under testing conditions. Depending on the chosen contrast criterion, the location of the best-focus surface can be slightly different. The best-focus surface becomes more curved in the presence of a planar refractive interface (cuvette wall and immersion medium) due to the presence of coma, astigmatism, and non-telecentric design of some objectives, which together bend the best-focus surface relative to the ideal conditions (air). On the contrary, the well-known Petzval field curvature does not depend on the presence of planar interfaces in the system, and it is independent of spherical aberration, coma, or astigmatism. The best-focus surface we measured in this work is similar to the concept of focal surface in optics.

## Discussion

The Benchtop mesoSPIM represents a major advancement in light-sheet microscopy for large cleared tissues, surpassing the previous mesoSPIM version in terms of optical performance, throughput, and range of possible applications. Its lower cost, ease of assembly, and

travel capability make it more affordable for research groups with a moderate budget. The modular design of the system allows for easy upgrades and maintenance, making it more future-proof than commercial systems. While some expertise is required for building and upgrading, it is within the reach of microscopy tech enthusiasts and does not require special skills.

In comparison to ExA-SPIM[30] and NODO[26] light-sheet microscopes, the Benchtop mesoSPIM offers lower resolution and throughput but also lower cost and higher versatility in the choice of detection objectives. On the other hand, compared to the descSPIM[57], the Benchtop mesoSPIM has a higher cost but also higher resolution, field of view, and throughput. This niche makes our system attractive to imaging facilities with a wide range of sample types and sizes and moderately high resolution and throughput requirements.

The objective testing method developed for this project enabled us to select optimal detection objectives for higher resolution and larger field of view. Although this method does not provide a quantification of PSF size, as other methods that employ fluorescent beads, it does not depend on the light-sheet illumination path, making it particularly useful for light-sheet systems with complex illumination techniques by decoupling detection properties from excitation. It can be also used for comparing objectives in other light microscopy modalities, and the resulting contrast maps are intuitive for interpretation. The method can be further improved by using several orientations of the test target, test targets of various line pair densities to probe the system MTF at multiple spatial frequencies, and by performing more advanced computational analysis of the contrast maps to deduce the optical aberrations present in the system.

The Benchtop mesoSPIM's axial and lateral resolution can be significantly improved by designing custom excitation and detection objectives that are corrected for the spherical aberration that comes from the medium mismatch. We thus encourage companies that design and produce microscopy objectives to develop long-working distance air objectives with NA 0.3–0.5, magnification 5×–20×, and built-in compensation of strong spherical aberrations (an equivalent of imaging through a glass block 25–35 mm thick). The ExA-SPIM[30] system circumvented this problem by using an industrial telecentric lens with the front beam splitter removed (an equivalent of 35 mm glass block), but this lens type is not infinity corrected, has a large footprint and a low magnification range, which limits its use in the emerging field of light-sheet imaging of cleared samples.

The Benchtop mesoSPIM can be made more compact through customized electronic components such as stage controller, waveform generator, power supplies, and laser combiner. The mechanical design can be further reconfigured to accommodate an off-the-shelve motorized turret with multiple detection objectives for higher user convenience. Although our assembly and some custom parts are licensed as open-source (GPLv3), we also use multiple closed-source components, and there is room for custom components from commercial manufacturers if the component's interface and performance are clearly specified and the design is not derived from already existing GPL-licenced parts.

Although we provide specific suggestions for the optomechanics, objectives, and other hardware for the Benchtop mesoSPIM, they are not exhaustive, and modifications can be made by other developers to further improve imaging quality, reduce cost, and adapt the system to numerous applications. We encourage open, test-based, and well-documented modifications to the system and its software, which can be shared with the imaging community through Github repositories.

We expect the Benchtop mesoSPIM and its modifications to have numerous applications in neuroscience, developmental biology, digital pathology, and physics.

## Methods

All procedures of our research were carried out according to all relevant ethical regulations. The mouse (iDISCO-cleared samples), tadpole, and chicken embryo experiments were conducted in accordance with standard ethical guidelines and the guidelines from the Veterinary Office of Switzerland and were approved by the Zurich Cantonal Veterinary Office. The mouse experiments for vDISCO-cleared samples followed European directive 2010/63/EU for animal research, reported according to the Animal Research: Reporting of In Vivo Experiments (ARRIVE) criteria, complied with the '3Rs' measure and were approved by the ethical review board of the government of Upper Bavaria (Regierung von Oberbayern, Munich, Germany) and conformed to institutional guidelines of Klinikum der Universität München/Ludwig Maximilian University of Munich). The human tissues were provided by the body donation program of the Department of Anatomy and Embryology, Maastricht University. The tissue donors gave their informed and written consent to the donation of their body for teaching and research purposes as regulated by the Dutch law for the use of human remains for scientific research and education ("Wet op de Lijkbezorging"). Accordingly, a handwritten and signed codicil from each donor posed when still alive and well, is kept at the Department of Anatomy and Embryology Faculty of Health, Medicine and Life Sciences, Maastricht University, Maastricht, The Netherlands.

### Optomechanical design

Continuing the mesoSPIM original design goals, we opted for long working distance air objectives with low NA (<0.3) for both detection and excitation, which ensures compatibility with all clearing methods and straightforward tiled imaging of large specimens.

The detection path can have either a *microscope* configuration (objective + tube lens) or a *telecentric lens* configuration (telecentric lens attached directly to the camera), see Fig. 1c and Supplementary Fig. 7. Custom-made lens and camera holders (red parts) are 3D printed from PETG or similar mechanically stable plastic. The objective-camera assembly is mounted on a focusing stage (ASI LS-50). The filter wheel is mounted either between the objective and the tube lens (infinity space), or in front of the telecentric lens.

The excitation path is simplified compared to mesoSPIM v.5 but has the same effective excitation of NA = 0.15. The left and right excitation arms are identical, each consisting of an optical fiber adapter, a collimator lens, an electrically tunable lens (ETL, Optotune EL-16-40-TC-VIS-5D-1-C), a 1:1 telescope (L2, L3, two Thorlabs AC254-100-A-ML), a galvo mirror (15 mm beam Thorlabs QS15X-AG), two folding mirrors,

and an excitation objective (L4, modified Nikon 50 mm f/1.4 G objective), see Supplementary Fig. 14 and Github Wiki (https://github.com/mesoSPIM/benchtop-hardware/wiki). In the excitation path, all 4 custom parts (folding mirror bracket, galvo mount, galvo heat sink, excitation objective mount, Fig. 1a, Supplementary Fig. 14, parts rendered in red) are machined from aluminum for thermal and mechanical stability. Switching of the laser between the arms is done at the laser combiner level (before the fiber coupling), with Oxxius L4Cc (405, 488, 561, 638 nm) laser combiner and with add-on module (product code "MDL-FSTM") for fast switching between the two laser fibers. The galvo assembly is shown in Supplementary Movie 12.

As in its predecessor, the sample immersion chamber (Supplementary Fig. 15) in the Benchtop mesoSPIM is stationary during image acquisition to ensure constant optical path for detection and excitation light. The immersion chamber (e.g., Portmann Instruments UG-753-H75 40×40×75 mm, fire-fused) is mounted on a kinematic table (Radiant Dyes RD-PDT-S) that allows tip, tilt and yaw adjustment of the chamber. The mounting is done with a custom-made chamber holder (3D printed from solvent-resistant plastic like PA12) and a magnetic kinematic base (Thorlabs KB25/M) that connects the holder with the kinematic table.

The sample is dipped into the immersion chamber and scanned through the light-sheet plane during the tile acquisition in a stepwise motion using the z-stage (ASI LS-50, 50 mm travel range with optical linear encoder and sub-micron accuracy). The vertical positioning between tiles is achieved with dual servo Y-stages (ASI LS-100 with 100 mm travel range), horizontal positioning with X-axis (ASI LS-50). Rotation of the sample is achieved with a rotary stage (ASI C60-3060-SRS).

The CAD design of mechanical components was performed in Autodesk Inventor 2023 software and the full model is available for download from the project repository.

### Vibration absorption

Unlike previous mesoSPIM versions that were mounted on an optical table with active vibration isolation, the Benchtop mesoSPIM is built on a rigid but light breadboard (Thorlabs UltraLight Series II Breadboard, #PBG52522) resting on four Ø27.0 mm sorbothane isolator feet (Thorlabs, # AV4/M). The mass of the microscope on the breadboard is 25 kg (6.25 kg/isolator), so according to the isolator specifications, vibration frequencies higher than 20 Hz are well-damped. To prevent low-frequency vibrations from people walking in the room, we recommend rooms with concrete floors and sturdy benches.

### Acquisition software

The user-friendly acquisition software[42] allows full control via GUI and configuration files. It is written using the PyQt5 platform with multithreading for high-performance imaging (e.g., frame grabbing and file writing run in parallel threads), and it remains the same for all mesoSPIM versions. The Benchtop-specific code is limited to the configuration file, which is individual for each system. A high-level control software block diagram and GUI windows are shown in Supplementary Fig. 8.

The electronics block diagram with the main computer-controlled components is shown in Supplementary Fig. 16. The waveform generation is identical to the mesoSPIM v.5 (Supplementary Fig. 3 in ref. 33), but the number of National Instruments DAQmx boards was reduced to one (PXI-6733 with one BNC-2110 connector block) to minimize the footprint and cost. The system supports up to 4 laser lines, with both analog and digital modulation.

The software currently supports saving in 16-bit RAW (binary), Fiji[44] TIFF, BigTIFF and Fiji BigDataViewer[45] H5/XML file formats, along with necessary metadata files. For multi-tile/channel/illumination acquisitions the Fiji BigDataViewer H5/XML format is preferred because of fast writing speed and saving of rich metadata in XML,

which makes the dataset ready for streamlined stitching and tiling in Fiji BigStitcher plugin[58], independently of its size and computer memory limitations.

## ASLM mode

The key feature of mesoSPIM which allows high and uniform z-resolution across large field of view is the axially swept light-sheet (ASLM) mode[33–36]. ASLM is achieved in Benchtop the same way as in mesoSPIM v.5, by using an ETL (Optotune EL-16-40-TC-VIS-5D-1-C) in each excitation arm. The optimal ETL parameters (offset and amplitude) are easily adjusted by the user in the acquisition software, using parked beam (galvo scanning stopped) and laser scattering in the imaging medium until a uniformly thin laser beam profile is achieved across the FOV in ASLM mode. This should be done in the absence of any sample in the FOV to avoid sample bleaching.

## Focus interpolation

For samples that are not well index-matched with the imaging medium or that do not use an immersion chamber, the focus distance depends on the imaging plane z-position in the sample. Moving the sample in z changes its detection path optical distance and makes it go out of focus. This issue is tackled in the acquisition software, where the user can define two focus position values (beginning and end of the stack), and the software linearly interpolates between them during stack acquisition.

## Stitching, fusion, and 3D visualization

All datasets were saved in Fiji BigDataViewer H5/XML format, stitched in BigStitcher plugin[58] and exported as either 16- or 8-bit TIFF files for visualization. The last column of Supplementary Table 3 shows the data size and the number of tiles for each sample. Movies and screenshots were made in Imaris software (Oxford Instruments).

## Detection objective testing: contrast and field flatness

Our goal was to design a general test of detection objective properties which would be independent of the light-sheet illumination, to allow fast and reliable screening of detection objectives. In detection objective properties we are concerned about the overall contrast at high spatial frequencies, the uniformity of contrast across the field, and the field flatness.

A high-contrast square wave Ronchi ruling target glass slide (76.2 mm × 25.4 mm) with 40 line pairs per mm (lp/mm) pattern (Thorlabs R1L3S14N) was immersed in BK7 matching liquid (RI 1.5167, Cargille Cat #: 19586) inside a tall glass chamber. The total thickness of glass and BK7 matching liquid was adjusted to 20 mm before the objective to mimic the imaging conditions of a cleared specimen. The Ronchi ruling was illuminated from the back using a smartphone (Motorola g8) with a white screen image to provide uniform, diffuse, incoherent illumination over a large FOV. The mesoSPIM camera exposure time was set to 50 ms to avoid temporal aliasing with the smartphone screen update rate.

During the acquisition, the chamber with Ronchi ruling was kept stationary, while the detection objective was moving in 10-μm steps, in a range from −300 to +300 μm from the best-focus position. Thus, a TIFF stack with 61 planes at various defocus positions was acquired. The analysis code splits each plane into a grid of sub-regions (32 ×20), and the contrast function is computed for each subregion:

$$C(X_i,Y_j,Z_k) = (Perc(99,subregion_{ijk}) - Perc(1,subregion_{ijk}))/$$
$$(Perc(99,subregion_{ijk}) + Perc(1,subregion_{ijk})) \quad (1)$$

where Perc() is the percentile function. Percentiles 1% and 99% were used instead of min() and max() respectively to account for noise.

The result is a 3D contrast map $C(x,y,z)$ which is an estimate of modulation transfer function (MTF) at a given spatial frequency (40 lp/mm) for a given detection objective. The best-focus surface was computed from $C(x,y,z)$ by taking its sections along $x$ and $y$, respectively, and measuring the sag (amplitude) of the high-intensity curve in each section (Supplementary Fig. 2). Note that $C(x,y,z)$ is affected by the residual tip/tilt of the Ronchi test target, which is taken into account in the field flatness calculation.

By taking the maximum of $C(x,y,z)$ along $z$ we also computed the maximum contrast map $C_{max}(x,y)$ of the objective (Supplementary Figs. 2–4, left plot in each panel). This metric is independent of the tip/tilt of the Ronchi test target and can be used to cross-compare detection objectives.

The contrast map $C(x,y,z)$ is also affected by the selection of tube lens, when infinity-corrected objectives are used. With Mitutoyo objectives, we use the standard MT-1 tube lens (f = 200 mm), which is designed to provide a FOV ("image circle" or "field number") up to 30 mm when placed within the nominal distance from the objective. With other objectives, we used an Olympus wide-field tube lens SWTLU-C (f = 180 mm, field number 26.5 mm). We noticed that mixing Mitutoyo objectives with SWTLU-C tube lens gave a flatter field, at a cost of 10% magnification reduction from nominal (by a factor 180/200 = 0.9).

## Chromatic effects on field flatness

To quantify the chromatic effects on detection objective contrast, we imaged the Ronchi slide (40 lp/mm) illuminated from behind (Supplementary Fig. 5b) with a HAL100 tungsten-halogen lamp (Zeiss) fitted with a set of bandpass filters and a diffuser. The Ronchi slide was imaged in the air for control, and then in a chamber with DBE (17 mm DBE + 3 mm of glass between the slide and the detection objective). The slide was moved along the z-axis at 10 μm steps between consecutive planes, thus emulating the conditions of a cleared tissue imaging.

The HAL100 lamp spectrum was balanced toward the blue range using a daylight blue filter (Olympus), which produced a more uniform spread of intensity across the visible spectrum (Supplementary Fig. 5b). Then, one of the following filters was applied to produce specific spectral bands: Chroma ET420/20, Semrock HQ470/30, Semrock Brightline HC 535/22, Semrock Brightline HC 595/31, Semrock Brightline Edge Basic 630/69 (combined with Edge Basic 594LP clean-up filter), Zeiss 697/75 (combined with Edge Basic 594LP). Lastly, the diffuser (120 grit, Thorlabs) was placed to ensure that light illuminates the Ronchi slide with a large angular spread. The illumination spectra were measured with the Hamamatsu VIS/NIR MiniSpectrometer model C10083CAH.

## Resolution measurement

Yellow-green fluorescent spheres (ex 441 nm, em 486 nm, Fluoresbrite® YG Microspheres 0.20 μm, Polysciences, Cat. # 17151) were mounted in 2% agarose in CUBIC-R+ medium (RI 1.52, TCI Chemicals, Prod. #T3741) inside a 10 × 10 × 45 mm glass cuvette. The sample cuvette was dipped in an immersion chamber filled with index-matched oil (RI 1.52, CUBIC mounting medium, TCI Chemicals, Prod. #M3294). Glass wall + oil thickness was 20 mm for Mitutoyo Plan Apo BD 5×/0.14 objective, and 5 mm for Mitutoyo 10×/0.28 and 20×/0.28 to control the spherical aberration effect but remain within realistic imaging conditions. The bead volume was acquired with 488-nm excitation laser and z-step = 1 μm.

From the resulting 3D stack of images, the bead centers were detected, filtered to exclude bead clusters, and bead image intensities were fitted with the Gaussian function along x−y plane and z-axis. The resulting PSF full width at half-maximum (FWHM) measurements were analyzed visually by color-coding the lateral resolution FWHM(x,y) and

axial FWHM(z) across FOV, and by looking at their histogram, mean and standard deviation.

As seen in Supplementary Fig. 1 histograms, the axial resolution of Benchtop is 3.3–4.0 μm at all magnifications (determined by the effective light-sheet thickness in ASLM mode, which is independent of the detection objective), and the lateral resolution ranges from 1.5 μm (Mitutoyo G Plan Apo 20×/0.28-t3.5) to 2.7 μm (Mitutoyo BD Plan Apo 5×/0.14), depending on the objective NA and the medium thickness used in the test.

## Sample mounting
iDISCO and ECi cleared samples were either placed in a dipping cuvette or clamped with adjustable 3D printed clamps (Fig. 1f) and dipped into an immersion chamber (Supplementary Fig. 15) containing DBE or ECi, respectively. The whole mouse body cleared with vDISCO (Fig. 5b) was placed into BABB-filled rectangular cuvette (Portmann Instruments UG-751-H80, 25×25×80mm) which was held on top by a metal clamp (Thorlabs BSH1/M) and imaged directly without dipping into immersion chamber, with focus interpolation.

All our sample holders were 3D printed from PA12 (polyamide) using industrial SLS printer at the UZH AMF BIOC facility.

## Video tutorials
We offer video tutorials at our channel: https://www.youtube.com/@mesoSPIM.

## iDISCO mouse brain clearing
All mouse experiments were conducted in accordance with standard ethical guidelines and the guidelines from the Veterinary Office of Switzerland and were approved by the Zurich Cantonal Veterinary Office.

Mice were maintained in small same-sex groups of 2–5 animals on a reversed 12/12 h light/dark cycle at relative humidity 50–60% and temperature 23 °C.

## Viral injection
For sparse retrograde viral labeling, we stereotactically injected an 8 months old male Vglut2-Cre mouse. The mouse was briefly anesthetized with isoflurane (2%) in oxygen in an anesthesia chamber and subsequently immobilized for intracerebral injection in a stereotactic frame (Kopf Instruments). Body temperature was constantly maintained at ~37 °C. To prevent drying, the mouse eyes were covered with vitamin A-containing cream (Bausch & Lomb). Anesthesia was maintained at 1% isoflurane in oxygen. We stereotactically injected ~200 nl of a retrograde AAV encoding a cre-dependent variant of td-Tomato (ssAAV-retro/2-CAG-dlox-tdTomato(rev)-dlox-WPRE-bGHp(A); concentration: 0.85×10E12 vg/ml; UZH Viral Vector Facility) in the right lateral habenula (LHb) using the following coordinates from bregma (in mm): −1.7 anteroposterior (AP), +0.5 mediolateral (ML), −2.5 dorsoventral (DV).

## Tissue labeling, collection, and clearing
Mouse brains were stained for different neuronal cell types or amyloid plaques and arterial vessels and cleared with a modified version of the iDISCO protocol[47]. For AAV-injected Vglut2-Cre mice, animals were used after 28 days of expression. For Thy1-positive neuronal labeling, B6.Cg-Tg(Thy1-YFP)16Jrs/J (#003709) female mouse, 10.8 months old was used. For plaque and arterial vessel labeling, B6.Cg-Tg(Thy1-APPSw,Thy1-PSEN1*L166P)21Jckr male mouse, 8 months old was used.

All mice were deeply anesthetized with ketamine (100 mg/kg body weight; Streuli Pharma AG) and transcardially perfused with ice-cold phosphate buffer saline (PBS), pH 7.4, followed by ice-cold 4% paraformaldehyde (PFA) in PBS. After 4.5 h of post-fixation at 4 °C shaking at 40 rpm, brains were washed with PBS for three times at room temperature (RT), with daily solution exchange.

Samples were dehydrated in serial incubations of 20%, 40%, 60%, 80% methanol (MeOH) in ddH$_2$O, followed by 2 times 100% MeOH, each for 1 h at RT and 40 rpm. Pre-clearing was performed in 33% MeOH in dichloromethane (DCM) overnight (o.n.) at RT and 40 rpm. After 2 times washing in 100% MeOH each for 1 h at RT and then 4 °C at 40 rpm, bleaching was performed in 5% hydrogen peroxide in MeOH for 20 h at 4 °C and 40 rpm. Samples were rehydrated in serial incubations of 80%, 60%, 40%, and 20% MeOH in ddH$_2$O, followed by PBS, each for 1 h at RT and 40 rpm. Permeabilization was performed by incubating the mouse brains 2 times in 0.2% TritonX-100 in PBS each for 1 h at RT and 40 rpm, followed by incubation in 0.2% TritonX-100 + 10% dimethyl sulfoxide (DMSO) + 2.3% glycine + 0.1% sodium azide (NaN3) in PBS for 3 days at 37 °C and 65 rpm. Blocking was performed in 0.2% Tween-20 + 0.1% heparine (10 mg/ml) + 5% DMSO + 6% donkey serum in PBS for 2 days at 37 °C and 65 rpm. All samples were stained gradually with the respective primary and secondary antibodies listed below in 0.2% Tween-20 + 0.1% heparine + 5% DMSO + 0.1% NaN3 in PBS (staining buffer) in a total volume of 1.5 ml per sample every week for 4 weeks at 37 °C and 65 rpm. Between first and secondary antibody incubation, samples were washed in staining buffer 4 times each for one hour followed by 5-th time o.n. at RT.

The following antibodies were used. For Vglut2-Cre mice, the primary polyclonal rabbit-anti-RFP antibody (Rockland, 600-401-379-RTU, dilution 1:2000) and secondary donkey-anti-rabbit-Cy3 antibody (Jackson ImmunoResearch, 711-165-152, dilution 1:2000) were used. For Thy1-YFP mice, the primary polyclonal chicken-anti-GFP antibody (Aves Labs, GFP-1020, dilution 1:400) and secondary donkey-anti-chicken-AlexaFluor594 antibody (Jackson Immuno Research, 703-585-155, dilution 1:400) were used. For APP/PS1 mice, the dye hFTAA[59] was used at dilution of 1:400 in combination with the Cy3-conjugated monoclonal anti-α-smooth muscle actin antibody (Sigma Aldrich, C6198) at dilution of 1:800.

After staining, mouse brains were washed in staining buffer 4 times each for one hour and 2 times o.n. at RT. Samples were dehydrated in serial incubations of 20%, 40%, 60%, 80% MeOH in ddH$_2$O, followed by 2 times 100% MeOH, each for 1 h at RT and 40 rpm. Clearing was performed in 33% MeOH in DCM o.n. at RT and 40 rpm, followed by incubation in 100% DCM 2 times each for 30 min. Refractive index matching (RI = 1.56) was achieved in dibenzylether (DBE).

## vDISCO mouse brain, spinal cord, and whole body clearing
Animal experiments of this part followed European directive 2010/63/EU for animal research, reported according to the Animal Research: Reporting of In Vivo Experiments (ARRIVE) criteria, complied with the '3Rs' measure and were approved by the ethical review board of the government of Upper Bavaria (Regierung von Oberbayern, Munich, Germany) and conformed to institutional guidelines of Klinikum der Universität München/Ludwig Maximilian University of Munich). The severity of the procedure was low. The animals were housed under a 12/12 h light/dark cycle. The ambient temperature was kept at 21 °C with a 45–65% of relative humidity.

The Thy1-GFP line M (2.5-month-old male) mouse, from which the brain and the spinal cord were dissected out, and the CX3CR1-GFP (P14, male) mouse were stained and cleared with vDISCO[60]. The detailed vDISCO protocol is available in ref. 21. Briefly, the 0.01 M PBS (1× PBS) perfused and 4% paraformaldehyde (Morphisto, 11762.01000) fixed mouse body was placed in a 300-ml glass chamber (Omnilab, 5163279) filled with 250–300 ml of solution, which covered the body completely. Next, a perfusion setting able to pump the solution in the glass chamber into the transcardial circulatory system of the mouse in recycling manner was established involving a peristaltic pump (ISMATEC, REGLO Digital MS-4/8 ISM 834; reference tubing, SC0266)

and two tubing channels: the first was set to circulate the solution through the heart into the vasculature by using a perfusion needle (Leica, 39471024), the second was immersed into the solution chamber where the animal was placed. To fix the needle tip in place and to ensure extensive perfusion, a drop of superglue (Pattex, PSK1C) was added onto the hole of the heart where the needle was inserted. By using this perfusion setting, the animal was first perfused with a decolorization solution for 2 days at room temperature, refreshing the solution when turned yellow, and then by a decalcification solution for 2 days at room temperature. In between these two solutions, a shorter step that consisted of perfusing the body with 1× PBS for 3 h 3 times at room temperature was performed as washing step. The decolorization solution was made with 25–30 vol% dilution of CUBIC reagent #1 (ref. 61): 25 wt% urea (Carl Roth, 3941.3), 25 wt% N,N,N′,N′-tetrakis (2-hydroxypropyl)ethylenediamine (Sigma-Aldrich, 122262) and 15 wt% Triton X-100 (AppliChem, A4975,1000) in 1× PBS. The decalcification solution consisted of 10 wt/vol% EDTA (Carl Roth, 1702922685) in 1× PBS, pH 8–9 adjusted with sodium hydroxide (Sigma-Aldrich, 71687).

Next, the mouse was perfused with 250 ml of permeabilization solution consisting of 1.5% goat serum (Gibco, 16210072), 0.5% Triton X-100, 0.5 mM of methyl−β−cyclodextrin (Sigma, 332615), 0.2% trans−1−acetyl−4−hydroxy−l−proline (Sigma-Aldrich, 441562) and 0.05% sodium azide (Sigma-Aldrich, 71290) in 1× PBS for half a day at room temperature. After this, the mouse was perfused for 6 days with 250 ml of the same permeabilization solution containing 290 µl of propidium iodide (PI, stock concentration 1 mg ml$^{-1}$ Sigma-Aldrich, P4864) and 35 µl of Atto647N conjugated anti-GFP nanobooster (Chromotek, gba647n-100). When the staining step was completed, the mouse was washed by perfusing with the washing solution (1.5% goat serum, 0.5% Triton X-100, 0.05% of sodium azide in 1× PBS) for 3 h 3 times at room temperature and with 1× PBS for 3 h 3 times at room temperature. Finally, the mouse was cleared in the glass chamber as follows: with passive incubation at room temperature in a gradient of tetrahydrofuran (Sigma-Aldrich, 186562)/double-distilled water (50, 70, 80, 100, and again 100 vol%) 12 h each step, followed by 3 h in dichloromethane (Sigma-Aldrich, 270997) and in the end in a mixture of benzyl alcohol (Sigma-Aldrich, 24122) and benzyl benzoate (Sigma-Aldrich, W213802) 1:2. During all incubation steps, the glass chamber was sealed with parafilm and covered with aluminum foil. All the clearing steps were performed in a fume hood.

### Human brain tissue preparation

The body donors were provided by the body donation program of the Department of Anatomy and Embryology, Maastricht University. The tissue donors gave their informed and written consent to the donation of their body for teaching and research purposes as regulated by the Dutch law for the use of human remains for scientific research and education ("Wet op de Lijkbezorging"). Accordingly, a handwritten and signed codicil from each donor posed when still alive and well, is kept at the Department of Anatomy and Embryology Faculty of Health, Medicine and Life Sciences, Maastricht University, Maastricht, The Netherlands. Three human occipital lobe samples were obtained from 2 body donors each (Occipital lobe 1: 101-year-old female; Occipital lobe 2: 90-year-old male; no known neurological disease, respectively). All six samples were taken approx. 3 cm anterior to the occipital pole and around the V1/V2 border (see Supplementary Figs. 9, 10). Brains were first fixed in situ by full-body perfusion via the femoral artery. Under a pressure of 0.2 bar the body was perfused by 10 l fixation fluid (1.8 vol% formaldehyde, 20% ethanol, 8.4% glycerin in water) within 1.5–2 h. Thereafter, the body was preserved at least 4 weeks for postfixation submersed in the same fluid. Subsequently, brains were recovered by calvarial dissection and stored in 4% paraformaldehyde in 0.1 M PBS until further processing.

The samples from Occipital lobe 1 were stained with MASH-MB and Occipital lobe 2 samples with MASH-NR[48], with minor adjustments

to the original protocol: the tissue was dehydrated in 20, 40, 60, 80, 100% MeOH in distilled water for 1 h each at RT, followed by 1 h in 100% MeOH and overnight bleaching in 5% H$_2$O$_2$ in MeOH at 4 °C. Samples were then rehydrated in 80, 60, 40, 20% MeOH, permeabilized 2× for 1 h in PBS containing 0.2% Triton X-100 (PBST). This was followed by a second bleaching step in freshly filtered 50% aqueous potassium disulfite solution. The samples were then thoroughly rinsed in distilled water 5× and washed for another 1 h. For staining, the samples were incubated in 0.001% methylene blue (Occipital lobe 1) or 0.001% neutral red (Occipital lobe 2) solution in McIlvain buffer (phosphate-citrate buffer)[62] at pH 4 for 5 days at room temperature. After 2.5 days the samples were flipped to allow for equal penetration of the dye from both sides. After staining, samples were washed 2× for 1 h in the same buffer solution, dehydrated in 20, 40, 60, 80, 2×100% MeOH for 1 h each, and delipidated in 66% dichloromethane (DCM)/33% MeOH overnight. This was followed by 2× 1 h washes in 100% DCM and immersion in ethyl cinnamate (ECi)[63].

### *Xenopus* tropicalis sample preparation

All tadpole experiments were conducted in accordance with standard ethical guidelines and the guidelines from the Veterinary Office of Switzerland and were approved by the Zurich Cantonal Veterinary Office.

Whole-mount *Xenopus* immunofluorescence was adapted from ref. 17. A stage 58 *Xenopus tropicalis* tadpole was fixed overnight in 4% PFA and dehydrated with 100% MeOH for 3 times 24 h. Sex of the animal was undetermined at the time of euthanisation, external characteristics for sex determination only become apparent at st. 62. Bleaching was performed under room lightning conditions in 10% H$_2$O$_2$ / 23% H$_2$O / 66% MeOH for 8 days, replacing the bleaching solution every 2 days until fully bleached.

The tadpole was stepwise rehydrated in 1× PBS with 0.1% Triton X-100 (PBST) and blocking was performed for 3 days at 4 °C in 10% CAS-Block / 90% PBST (008120, Life Technologies). Staining was performed using Atp1a1 antibody (1:500, DSHB, a5) diluted in 100% CAS-Block for 5 days at 4 °C on a rotating wheel. The tadpole was washed 3 times for 8 h and 2 times for 1 day with PBST at RT, blocked again for 1 day at 4 °C (10% CAS-Block / 90% PBST) and incubated 72 h at 4 °C on a rotating wheel with the secondary antibody (1:500, Alexa-Fluor-555, 405324 P4U/BioLegend UK Ltd) diluted in 100% CAS-Block. The tadpole was washed for 3 × 8 h and 2×1 day with PBST.

The tadpole was dehydrated in 25% MeOH/75% 1× PBS (1 day), 50% MeOH/50% 1× PBS (1 day), 75% MeOH/25% 1× PBS (1 day), three times 100% MeOH (2 times 8 h, 1 time 24 h). Finally, clearing was performed in BABB (benzyl alcohol:benzyl benzoate 1:2, (Merck 8187011000 and Sigma 108006) for 48 h.

For samples imaged on the SPIM-tower, BMS-453-treated (1 µM final concentration, 0.1% DMSO) and untreated embryos (0.1% DMSO) were fixed at respectively st. 33, 37 and 42, stained and cleared following the protocol in ref. 17. In brief, the primary antibody mix was anti-Atp1a1 (1:200, DSHB, A5), anti-col2a1 (1:200, DSHB, II-II6B3) and anti-TNNT2 (1:200, DSHB, CT3). The secondary antibody was Alexa Fluor 555 goat anti-mouse IgG (minimal x-reactivity) (405324, P4U/BioLegend UK). For nuclear counterstaining, DAPI (20 µg/ml, ThermoFisher, D1306) was added to the primary antibody mixture.

### Chicken embryo

All chicken embryo experiments were conducted in accordance with standard ethical guidelines and the guidelines from the Veterinary Office of Switzerland and were approved by the Zurich Cantonal Veterinary Office.

Neurofilament staining of a whole-mount chicken embryo: the embryo was sacrificed at day 9 of development and fixed in 4% paraformaldehyde for 3.5 h at room temperature. The sex of the embryo could not be determined at this stage. For best results, the embryo was

kept in constant, gentle motion throughout the staining procedure. Incubations were at 4 °C if not declared differently. The tissue was permeabilized in 1% Triton X-100/PBS for 30 h, followed by an incubation in 20 mM lysine in 0.1 M sodium phosphate, pH 7.3 for 18 h. The embryo was rinsed with five changes of PBS containing 0.2% Triton. To avoid unspecific antibody binding the embryo was incubated in 10% FCS (fetal calf serum) in PBS for 48 h. The primary antibody mouse anti neurofilament (1:1500, RMO270, Thermofisher Scientific, Cat-nr. 13-0700) was added for 60 h. Unbound primary antibody was removed by ten changes of 0.2% Triton/PBS and an additional incubation overnight. After re-blocking in 10% FCS/PBS for 48 h, the embryo was incubated with the secondary antibody goat anti-mouse IgG-Cy3 (1:1'000, Jackson ImmunoResearch 115-165-003) for 48 h. Afterwards, the embryo was washed ten times with 0.2% Triton/PBS followed by incubation overnight. For imaging, the tissue was dehydrated in a methanol gradient (25%, 50%, 75% in $H_2O$ and 2×100%, minimum 4 h each step at room temperature) and cleared using 1:2 benzyl alcohol: benzyl benzoate (BABB) solution (again gentle shaking is recommended for dehydration and clearing). The tissue and staining are stable for months when kept at 4 °C in the dark.

## CaF₂ crystal preparation

Four 1 cm³ $CaF_2$ crystals, all sides polished, were acquired from Crystran. The shape and polish of the crystals were chosen to minimize multiple scattering of light in the sample. Two of these crystals were irradiated in a $^{60}$Co irradiator at Penn State University. The crystals received dosages of ~100 kRad and 5 MRad each. As the radiation source was isotropic and the attenuation of 1.17 MeV and 1.33 MeV gamma-rays by 1 cm of $CaF_2$ is low[64], the deposition of energy by the gamma-rays in the crystal is expected to be uniform. The two other crystals were left without irradiation for comparison (blank references).

The crystals were placed in a custom-made sample holder (Supplementary Fig. 12a, b) and were imaged without immersion, both before and after gamma-ray irradiation. By comparing the images before and after irradiation (as well as irradiated vs blank), we demonstrate that the mesoSPIM can clearly measure fluorescence from color centers induced by irradiation. Figure 6b shows the images in response to the 405 nm laser. The response at different laser settings, comparison to blank, and distribution of color centers are shown in Supplementary Fig. 12 and discussed in Supplementary Note 2.

## Statistics and reproducibility

For the application examples (Figs. 2, 4, and 5a–l), no statistical method was used to predetermine sample size, no data were excluded from the analyses, the experiments were not randomized, and the investigators were not blinded to allocation during experiments and outcome assessment. These choices were made because these were demo/pilot samples and testing specific biological hypotheses required additional experiments. The resolution measurements (Supplementary Fig. 1) and field flatness analysis (Supplementary Fig. 6) have inherently negligible inter-sample variability because they use fixed calibration samples (fluorescent microspheres and Ronchi slides).

For the color center imaging (Fig. 6a–c, Supplementary Fig. 12), four CaF₂ crystals were measured, with two crystals scanned both before and after irradiation, while two crystals remained in an unirradiated (blank) state as control samples. For each dataset, between 500 and 1000 images were acquired at 10 μm z-steps. Scans of irradiated crystals were performed twice consecutively to generate an additional dataset used for further validation of signal versus noise. Background levels were estimated by measuring the camera noise while the laser was off and by measuring the blank control samples. Fluorescence signals from color centers were extracted from the datasets of irradiated crystals. To derive background and signal

intensity distributions and averages, 350 images from each dataset were utilized. The resulting averages, along with their standard deviations, are presented in Supplementary Fig. 12f.

To identify and validate the presence of fluorescent structures within the uniformly illuminated volume of the crystals, regions containing clusters of pixels with non-zero intensity were processed using the feature.match_template method of the scikit-image library[65]. This method provided cross-correlation values between identified clusters ('structures') and volumes around the corresponding regions in the repeated scans. As illustrated in Supplementary Fig. 12d and described in Supplementary Note 2, the resulting distributions of cross-correlation values exhibited either a purely random profile, resembling a Gaussian curve with a mean value of zero, or a similar distribution with an outlier point displaying high correlation (7–8 sigma away from the mean). Examples of the latter, along with further details on the analysis, are provided in Supplementary Fig. 12d and Supplementary Note 2.

## Reporting summary

Further information on research design is available in the Nature Portfolio Reporting Summary linked to this article.

## Data availability

The raw and processed data generated in this study for Figs. 2, 4 and 5a, Supplementary Figs. 1, 6 have been deposited in the BioImage Archive under accession number S-BIAD963. Source data are provided in this paper.

## Code availability

The mesoSPIM control software[42] (v.1.8.3): https://github.com/mesoSPIM/mesoSPIM-control The Benchtop CAD design files, list of parts, and building instructions (Wiki): https://github.com/mesoSPIM/benchtop-hardware The detection objective testing, including chromatic effects on field flatness: https://github.com/nvladimus/lens-testing The PSF quantification (ETL-on and -off cases): https://github.com/mesoSPIM/mesoSPIM-PSFanalysis The optical simulations in Optalix software: https://github.com/mesoSPIM/optical-simulations

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

## Acknowledgements

The authors thank Stefan Giger from the Brain Research Institute for mechanical prototyping, Fabian Eggiman, Lukas Lüchinger, and Sascha Weidner from the Additive Manufacturing Facility (AMF) of the University of Zurich (UZH) for help with designing and 3D printing of custom sample holders, Daniel Invernot, Melanie Horn and Julia Traversari from UZH for *Xenopus* animal care and husbandry, Andrew Woehler from HHMI Janelia JET for providing Thorlabs super apochromat objectives, Giulia Miracca from the University Hospital Zurich for requesting and testing BSL-3 cuvette clamps; Igor Jovanovic from the University of Michigan for the irradiation of crystals, Michel Rickhaus and Joseph Woods from UZH for the use of the fluorescence spectrophotometer, and Celine Heeb and Lorenz Weber for help with Wiki instructions on building a Benchtop mesoSPIM. This work was supported by the University Research Priority Program (URPP) "Adaptive Brain Circuits in Development and Learning (AdaBD)" of the University of Zurich (N.V., E.S. and F.H.). In addition, F.F.V. is supported by an HFSP fellowship (LT00687), T.N. received funding from H2020 Marie Skłodowska-Curie Actions (xenCAKUT - 891127), A.R. and S.H. were supported by a Dutch Science Foundation VIDI Grant (14637), and A.R. was supported by an ERC Starting Grant (MULTICONNECT, 639938). Further funding support came from the Swiss National Science Foundation (SNF grant nos. 31003B-170269, 310030_192617 and CRSII5-18O316 to F.H., 310030_189102 to S.S.L.), 200020_204950 to L.B., G.R.A., and V.A.); from an ERC Starting Grant by the European Union's Horizon 2020 Research and Innovation Program (grant agreement no. 804474, DiRECT, S.S.L); and the US Brain Initiative (1U01NS090475-01, F.H.).

## Author contributions

Project conceptualization: N.V., F.F.V., and F.H. Optomechanical and electronics design: N.V., and F.F.V. Objective testing method: N.V. CAD design: N.V., F.F.V., G.R.A., and P.S. Software development: N.V., and F.F.V. Experiments conceptualization: T.N., G.R.A., L.B., S.S.L. Imaging and data collection: N.V., F.F.V., T.N., and G.R.A. Data analysis and/or visualization: N.V., F.F.V., T.N., G.R.A., and V.A. Sample preparation: T.N., G.R.A., R.C., A.M.R., S.Z., S.H., M.S., D.G., and T.Y. Writing the manuscript: N.V., F.F.V., T.N., G.R.A., R.C., A.M.R., S.Z., S.H., M.S., D.G., L.B., S.S.L., and F.H. Editing the manuscript: N.V., F.F.V., T.N., G.R.A., R.C., A.M.R., S.Z., S.H., M.S., D.G., J.M.M., P.B., L.B., A.R., E.S., S.S.L., and F.H. Funding acquisition: F.F.V., T.N., A.R., A.E., A.A., U.Z., E.S., L.B., S.S.L., F.H. Supervision, resources: A.R., A.E., A.A., U.Z., E.S., L.B., S.S.L., and F.H.

## Competing interests

The authors declare no competing interests.
