## [Peer Review File · Nature Communications]

Benchtop mesoSPIM: a next-generation open-source light-sheet microscope for cleared samplesREVIEWERS' COMMENTS

Reviewer #1 (Remarks to the Author)

This manuscript by Vladimirov, Helmchen and colleagues proposes their latest version of mesoSPIM microscopy named the Benchtop mesoSPIM. Comparing with their initial version (Voigt et al. Nature Methods 2019), This version includes several significant updates in features, a larger CMOS sensor, an optimized detection system, and a size reduction, resulting in higher resolution imaging with the benchtop-sized devise. The authors also developed a new, easier method to evaluate microscopy's objective properties. Based on the results, they finally selected Mitutoyo's Plan Apo BD/G series for their new setup. The authors obtained multiple representative images of whole-organ and whole-body cleared samples, as well as color senters in CaF2 crystals that particularly intrigued this reviewer.

Overall, the microscopy specialists' manuscript does a decent job of describing the new features and background development processes of the modified system. This reviewer was almost satisfied with the current manuscript, although a few points to be amended were found.

#1 In the 4th paragraph of "Results":

There is insufficient data to demonstrate ALL clearing methods' compatibility. LSFM typically necessitates high sample transparency, whereas a portion of clearing methods do not necessarily pursue high sample transparency, as each clearing protocol has various objectives (e.g., protocols with enhanced preservation of biological membranes). Considering the variety of tissue clearing protocols, this reviewer believes that the description should be modified to be more moderate.

#2 In the 4th paragraph of "Testing of microscope objective contrast properties":

"which created a high index medium..." What index was argued here?

#3 In the 6th paragraph of "Testing of microscope objective contrast properties":

As the authors mentioned, multi-color imaging applications are often required in biological research. The chromatic aberrations of Mitsutoyo's objective lenses tested on a mesoSPIM setup and compared to life science microscopy objectives may be useful information for users and readers. Is it possible to additionally provide this type of evaluation data?

#4 In the "Imaging of color centers for particle detectors":

It is so interesting that mesoSPIM enables physical studies by visualizing color centers. Color centers also assess the crystal structure of semiconductors in connection with such applications. For example, the RI of GaN solid crystal is ca. 2.4, which is much higher than that of cleared biological tissues. As per the reviewer's accuracy, is mesoSPIM applicable to such a high RI environment? I suppose that showing the maximum range of applicable RIs is useful not only for biologists but also for applied physicists and others.

#5 Methods

Regarding the sample handling, this reviewer supposed that a specific care for organic solvent-based clearing would be required for putting the sample into the dipping cuvette and immersing the

sample chamber. Is it possible to add more details? Can the dipping cuvette completely seal the cleared sample with the reagent? How was the evaporation of the reagent in the sample chamber avoided or prevented during imaging?

#6 methods

Regarding the iDISCO procedure, the dilution rates of primary and secondary antibodies were not described explicitly. Each dilution rate should be elaborated as in the other immunostaining part.

Minor comments:

- The common name "Thy1-GFP line M" would be better instead of "Thy1-GFPM"

<https://www.jax.org/strain/007788>

- Fig 1d: This reviewer suggests other colors instead of blue and green; they are not necessarily appropriate to distinguish.

Reviewer #2 (Remarks to the Author)

(co-reviewed with Reviewer #1)

Reviewer #3 (Remarks to the Author)

The authors of the paper introduce their next-generation open-source microscope, which they refer to as mesoSPIM (Benchtop). The primary enhancement in the "Benchtop" version is the utilization of flat-field (or field curvature) -corrected objectives, in contrast to the previous employment of MVPLAO objectives with a zoom body, accompanied by the integration of a new camera.

On one hand, the technical advancements here is incremental; however, on the other hand, the authors excel in democratizing science. They have remarkably enabled researchers with constrained resources to access a light-sheet microscope costing less than \$100k. For instance, the authors actively maintain an up-to-date website, provide a software package for operating the microscope, and provide tutorial videos on microscope assembly and related procedures. The authors also present evidence of multiple institutions in the US and Europe adopting their system, underscoring the significance of their work.

In general the work support the claims, but several key comments could contribute to improving the manuscript:

1. The introduction section appears lengthy and convoluted. The authors should succinctly state the gaps and their proposed solutions from the outset.
2. It would be valuable for the authors to include a section elucidating how they achieve an axial resolution of 3.3 μm for a 0.14 NA objective lens. Many readers are familiar with the depth-of-field formula ($\sim \lambda / \text{NA}^2$). An explanation how the axial resolution is determined in light sheet microscopy is required as presented in doi:10.1038/nprot.2014.087, which provides the depth-of-field formula in relation to excitation and detection objectives. This section should also address the role of axial scanning.
3. While the use of a flat-field-corrected objective promises improved outcomes, the authors

should address why their previous design was acceptable despite lacking this correction.

4. To comprehensively evaluate the axial resolution claims across the field of view, inclusion of the excitation beam profile is essential. Given the presence of numerous glass-air-liquid interfaces, it is reasonable to anticipate spherical aberrations that broaden the focal spot. The authors should discuss this effect on performance, and how the different mounting options affect spherical aberrations.

5. Data showcasing the co-localization of different colors across the large field of view is crucial. Significant chromatic aberrations could arise over a large field of view, potentially affecting dual or triple staining experiments. Users should be conscious of these microscope limitations.

6. The approach to resolving vibration issues merits elucidation. Clarification is needed on whether this is unique to the setup or if the tested locations possess minimal vibrations.

7. When reporting imaging times, such as 13 minutes, details should be provided regarding magnification, axial sampling, and whether image capture occurs during stage movement. Elucidating the impact of capturing images while the stage is in motion on registration, blurring and related factors is important.

8. A paragraph dedicated to limitations and future research is clearly absent from the manuscript. Addressing these limitations would contribute to a more comprehensive presentation.

RESPONSE TO REVIEWERS' COMMENTS

We thank the reviewers for their comments and helpful suggestions. Find below our point-by-point responses to their points.

Responses to Reviewer #1

This manuscript by Vladimirov, Helmchen and colleagues proposes their latest version of mesoSPIM microscopy named the Benchtop mesoSPIM. Comparing with their initial version (Voigt et al. Nature Methods 2019), This version includes several significant updates in features, a larger CMOS sensor, an optimized detection system, and a size reduction, resulting in higher resolution imaging with the benchtop-sized device. The authors also developed a new, easier method to evaluate microscopy's objective properties. Based on the results, they finally selected Mitutoyo's Plan Apo BD/G series for their new setup. The authors obtained multiple representative images of whole-organ and whole-body cleared samples, as well as color centers in CaF₂ crystals that particularly intrigued this reviewer.

Overall, the microscopy specialists' manuscript does a decent job of describing the new features and background development processes of the modified system. This reviewer was almost satisfied with the current manuscript, although a few points to be amended were found.

Reply: We thank the reviewer for appreciating our work and for his/her useful feedback.

#1 In the 4th paragraph of "Results":

There is insufficient data to demonstrate ALL clearing methods' compatibility. LSFM typically necessitates high sample transparency, whereas a portion of clearing methods do not necessarily pursue high sample transparency, as each clearing protocol has various objectives (e.g., protocols with enhanced preservation of biological membranes). Considering the variety of tissue clearing protocols, this reviewer believes that the description should be modified to be more moderate.

Reply: Thank you, we changed the text accordingly: “all” changed to “multiple”.

#2 In the 4th paragraph of "Testing of microscope objective contrast properties":

"which created a high index medium..." What index was argued here?

Reply: We mean the index of 1.52 as stated in the text:

"The air objectives were tested in two conditions: first, with the Ronchi grating in air, and then with the Ronchi grating immersed in oil with a refractive index of 1.52, which created a high-index medium approximately 20 mm thick between the objective and the sample (as shown in Fig. 2a)."

For clarity we changed this sentence to

"The air objectives were tested in two conditions: first, with the Ronchi grating in air, and then with the Ronchi grating immersed in oil with a refractive index of 1.52, approximately 20 mm

thick between the objective and the sample (as shown in Fig. 2a)."

#3 In the 6th paragraph of "Testing of microscope objective contrast properties":
As the authors mentioned, multi-color imaging applications are often required in biological research. The chromatic aberrations of Mitutoyo's objective lenses tested on a mesoSPIM setup and compared to life science microscopy objectives may be useful information for users and readers. Is it possible to additionally provide this type of evaluation data?

Reply: We thank all reviewers for their thoughtful questions about chromatic aberrations. To answer these questions, we performed additional experiments with 6 chromatically defined channels of illumination and analyzed the data. We found significant chromatic shifts of best-focus positions in all experiments, and a weak dependence of objective field curvature on the wavelength. We show these results in the new SI Figs. 5 and 6 and we added the following paragraph at the end of the Results section „Testing of microscope objective contrast properties“, p.4-5:

“The testing method described above uses white light for Ronchi slide illumination, where the smartphone screen emits with red, green and blue LEDs (emission spectrum shown in SI Fig 5a). This restriction limits quantification of chromatic effects that would be representative for fluorescence channels. To quantify the role of chromatic effects we therefore expanded our testing method with a tungsten-halogen lamp and a set of 6 bandpass filters (spectra shown in SI Fig.5b), and a Ronchi slide immersed in DBE (20 mm medium before the slide).

For a control, we imaged the Ronchi slide in the air using the Olympus MVPLAPO-1x objective at 2x zoom of MVX-10 body and found relatively flat field at all chromatic channels (SI Fig 6a), indicating that the objective performed nominally. This was in stark contrast with our earlier measurements when Ronchi slide was immersed in oil and illuminated with white light (SI Fig. 3). Indeed, we could reproduce the strongly curved field again when the slide was immersed in DBE (SI Fig 6b), which indicates that best-focus surface becomes curved by the presence immersion medium itself. This presumably occurs because the objective is not telecentric at low zoom, so the combination of spherical aberration, coma and astigmatism distort the best-focus field and render it non-flat. Additionally, large chromatic focal shifts are visible with the DBE-immersed Ronchi slide. Notably, focus offsets relative to the shortest wavelength (420 nm) were positive in DBE but negative in air, presumably due to DBE chromatic dispersion.

We found that under our emulated “cleared tissue” conditions, both the Olympus MVPLAPO-1x objective and the Mitutoyo objectives have significant chromatic focal offsets (i.e. best-focus planes vary for different channels), up to 400 μm between blue (420/20 filter) and red (697/75 filter) channels (SI Fig. 6b-e). For example, the focal plane offset between GFP (535/22 filter) and RFP (630/69 filter) channels can be between 50 and 150 μm , positive or negative, depending on the objective. In most cases the dependence of best-focus field profile on chromatic channel was rather weak (except MVPLAPO-1x at zoom 2x), which suggests that channel-dependent focus offset is the main factor to consider for optimal imaging performance.

In practical terms, the mesoSPIM detection objective must be focused differently for each channel, which is achieved through the control software (Focus button group, SI Fig.8). The amount of refocus depends on the channel, medium, and objective, and is adjusted manually for each specific set of conditions.”

The newly added sections are highlighted with **turquoise in the revised manuscript.**

#4 In the "Imaging of color centers for particle detectors":

It is so interesting that mesoSPIM enables physical studies by visualizing color centers. Color centers also assess the crystal structure of semiconductors in connection with such applications. For example, the RI of GaN solid crystal is ca. 2.4, which is much higher than that of cleared biological tissues. As per the reviewer's accuracy, is mesoSPIM applicable to such a high RI environment? I suppose that showing the maximum range of applicable RIs is useful not only for biologists but also for applied physicists and others.

Reply: This is an interesting question. We simulated the detection path with medium refractive index up to 1.99 (the limit that our simulation software allows) and found only a weak dependence of resolution on immersion medium refractive index. Therefore, we predict that imaging of crystals with RI > 2.0 is also feasible. We added these simulations in the new SI Note 4 and SI Table 11.

#5 Methods

Regarding the sample handling, this reviewer supposed that a specific care for organic solvent-based clearing would be required for putting the sample into the dipping cuvette and immersing the sample chamber.

Is it possible to add more details?

Reply: The details depend on the specific clearing protocol, and we have Wiki pages dedicated to several popular protocols: https://github.com/mesoSPIM/mesoSPIM-hardware-documentation/wiki/mesoSPIM_sample_handling

Can the dipping cuvette completely seal the cleared sample with the reagent?

Reply: Yes, the dipping cuvettes come with a Teflon cap which can be sealed to the cuvette, for example with silicone sealant. For non-solvent protocols (CLARITY, CUBIC), the cuvette can also be sealed with parafilm.

How was the evaporation of the reagent in the sample chamber avoided or prevented during imaging?

Reply: BABB, DBE and ECi evaporate rather slowly, so evaporation is generally not an issue. These details are beyond the scope of our current paper, but we will make effort to add them to the wiki and to our YouTube channel in the near future.

#6 methods

Regarding the iDISCO procedure, the dilution rates of primary and secondary antibodies were not described explicitly. Each dilution rate should be elaborated as in the other immunostaining part.

Reply: Thank you, we amended the iDISCO details in the revised manuscript as follows:

"All samples were stained gradually with the respective primary and secondary antibodies listed below in 0.2% Tween-20 + 0.1% heparine + 5% DMSO + 0.1% NaN₃ in PBS (staining buffer) in a total volume of 1.5 ml per sample every week for 4 weeks at 37°C and 65 rpm.

Between first and secondary antibody incubation, samples were washed in staining buffer 4 times each for one hour followed by 5-th time o.n. at RT.

The following antibodies were used. For Vglut2-Cre mice, the primary polyclonal rabbit-anti-RFP antibody (Rockland, 600-401-379-RTU, dilution 1:2000) and secondary donkey-anti-rabbit-Cy3 antibody (Jackson ImmunoResearch, 711-165-152, dilution 1:2000) were used. For Thy1-YFP mice, the primary polyclonal chicken-anti-GFP antibody (Aves Labs, GFP-1020, dilution 1:400) and secondary donkey-anti-chicken-AlexaFluor594 antibody (Jackson Immuno Research, 703-585-155, dilution 1:400) were used."

Minor comments:

- The common name "Thy1-GFP line M" would be better instead of "Thy1-GFPM"

<https://www.jax.org/strain/007788>

Reply: We changed it, thank you.

- Fig 1d: This reviewer suggests other colors instead of blue and green; they are not necessarily appropriate to distinguish.

Reply: Thank you, we changed this to blue and orange, following the recommendations in <https://davidmathlogic.com/colorblind/>.

Reviewer #2

(co-reviewed with Reviewer #1)

Reviewer #3

The authors of the paper introduce their next-generation open-source microscope, which they refer to as mesoSPIM (Benchtop). The primary enhancement in the "Benchtop" version is the utilization of flat-field (or field curvature) -corrected objectives, in contrast to the previous employment of MVPLAO objectives with a zoom body, accompanied by the integration of a new camera.

On one hand, the technical advancements here is incremental; however, on the other hand, the authors excel in democratizing science. They have remarkably enabled researchers with constrained resources to access a light-sheet microscope costing less than \$100k. For instance, the authors actively maintain an up-to-date website, provide a software package for operating the microscope, and provide tutorial videos on microscope assembly and related procedures. The authors also present evidence of multiple institutions in the US and Europe adopting their system, underscoring the significance of their work.

Reply: We thank the reviewer for his/her comments and for highlighting the significance of our work.

In general the work support the claims, but several key comments could contribute to improving the manuscript:

1. The introduction section appears lengthy and convoluted. The authors should succinctly state the gaps and their proposed solutions from the outset.

Reply: We rewrote the entire Introduction, trimmed unnecessary sentences, and articulated the gaps in the current state of the art that we are aiming to fill with our manuscript. We hope this significantly improved and streamlined the Introduction.

2. It would be valuable for the authors to include a section elucidating how they achieve an axial resolution of 3.3 μm for a 0.14 NA objective lens. Many readers are familiar with the depth-of-field formula ($\sim \text{Lambda}/\text{NA}^2$). An explanation how the axial resolution is determined in light sheet microscopy is required as presented in doi:10.1038/nprot.2014.087, which provides the depth-of-field formula in relation to excitation and detection objectives. This section should also address the role of axial scanning.

Reply: The explanation of the formula for axial resolution in light-sheet microscopy goes beyond the scope of present paper, there are excellent reviews on this topic, such as Power and Huisken, A guide to light-sheet fluorescence microscopy for multiscale imaging, Nature Methods, 2017, doi:10.1038/nmeth.4224. We briefly explain the axial scanning concept in the Introduction, and have added the reference to Power and Huisken 2017 paper, on p.2:

“The mesoSPIM system achieves uniform axial resolution across cm-scale FOV by using the axially swept light-sheet microscopy principle (ASLM)^{21,23–25}. In brief, the axially most confined region of the light-sheet (the excitation beam waist) is moved through the sample in synchrony with the camera’s programmable rolling shutter by using an electrically tunable lens (ETL) as remote focusing device. The synchrony of the beam translation and the camera readout leads to uniform axial resolution across the field of view (see reviews in Refs^{25,26}).”

3. While the use of a flat-field-corrected objective promises improved outcomes, the authors should address why their previous design was acceptable despite lacking this correction.

Reply: Our previous design was also based on a plan apochromat objective (Olympus MVPLAPO-1x). In our manuscript we show that the performance of this objective depends zoom setting of the microscope body (MVX-10) and the presence of imaging medium, with low zoom settings (1.25 and 2x) performing poorly. However, many impressive datasets were acquired with that system at higher zoom settings (4x, 5x, and 6.3x), at which the objective performs much better in terms of contrast uniformity and field flatness. Only in this manuscript we demonstrate how to measure and compare the desired qualities of the detection objectives and how to make informed decisions for future designs. We did not imply that the previous design is unacceptable for real-world applications. We have clarified this in the following sentences in Results section, p. 4 (newly added parts are underscored):

“To our surprise, the contrast maps of the Olympus MVPLAPO-1x objective with zoom body MVX-10 (the default mesoSPIM v.5 configuration) showed poor centricity and large variation of contrast across field at some zoom settings, especially at 1.25x and 2x zoom (Fig. 2c, right panels, see further SI Fig. 3), presumably because of the aberrations induced by the immersion medium (see below).”

and further, p. 5:

“For a control, we imaged the Ronchi slide in the air using the Olympus MVPLAPO-1x objective at 2x zoom of MVX-10 body and found relatively flat field at all chromatic channels

(SI Fig 6a), indicating that the objective performed nominally. This was in stark contrast with our earlier measurements when Ronchi slide was immersed in oil and illuminated with white light (SI Fig. 3). Indeed, we could reproduce the strongly curved field again when the slide was immersed in DBE (SI Fig 6b), which indicates that best-focus surface becomes curved by the presence immersion medium itself.”

4. To comprehensively evaluate the axial resolution claims across the field of view, inclusion of the excitation beam profile is essential. Given the presence of numerous glass-air-liquid interfaces, it is reasonable to anticipate spherical aberrations that broaden the focal spot. The authors should discuss this effect on performance, and how the different mounting options affect spherical aberrations.

Reply: We thank the reviewer for this question, this information was indeed missing from the manuscript text. To address this issue, we imaged fluorescent beads with 488-nm laser excitation and the ETL mode off and quantified the FWHM of the axial PSF, which is a measure of the beam waist profile. The results yielded a mean waist profile of $2.8 \pm 0.33 \mu\text{m}$ (\pm s.d.; $n = 103$). Detailed analysis is available at our public repository [https://github.com/mesoSPIM/mesoSPIM-PSFanalysis/benchtop-MitutoyoBD-5X\(TL-MT1\)-ETL-off.ipynb](https://github.com/mesoSPIM/mesoSPIM-PSFanalysis/benchtop-MitutoyoBD-5X(TL-MT1)-ETL-off.ipynb)

The measured beam waist is indeed thicker than one would expect from a diffraction-limited 488 nm, 0.15 NA Gaussian beam ($0.61 \cdot 0.488 / 0.15 = 2.0 \mu\text{m}$), due to spherical aberration from the imaging medium in the cuvette, imperfection of the excitation objective, or both.

We have extensively simulated the effect of spherical aberration (due to the immersion medium) on excitation beam diameter and added a new *Supplementary Note 3* with the results of these measurements, simulations, and our interpretations.

5. Data showcasing the co-localization of different colors across the large field of view is crucial. Significant chromatic aberrations could arise over a large field of view, potentially affecting dual or triple staining experiments. Users should be conscious of these microscope limitations.

Reply: This is a great question, please see our response to Reviewer 1, point #3, on the effect of chromatic effects.

6. The approach to resolving vibration issues merits elucidation. Clarification is needed on whether this is unique to the setup or if the tested locations possess minimal vibrations.

Reply: We updated Methods with the following paragraph in Methods, p.10:

“Vibration absorption

Unlike previous mesoSPIM versions that were mounted on an air-floating optical table, the Benchtop mesoSPIM is built on a rigid but light breadboard (Thorlabs UltraLight Series II Breadboard, #PBG52522) resting on four $\text{\O}27.0$ mm sorbothane isolator feet (Thorlabs, #AV4/M). The mass of the microscope on the breadboard is 25 kg (6.25 kg/isolator), so according to the isolator specifications, vibration frequencies higher than 20 Hz are well damped. To prevent low-frequency vibrations from people walking in the room, we recommend rooms with concrete floors and sturdy benches.”

7. When reporting imaging times, such as 13 minutes, details should be provided regarding magnification, axial sampling, and whether image capture occurs during stage movement. Elucidating the impact of capturing images while the stage is in motion on registration, blurring and related factors is important.

Reply: We have reported the imaging times dependence on magnification and axial sampling in the *Supplementary Notes, SI Table 8*.

Regarding the stage movement during acquisition, we have already mentioned these details in the *Methods, p.10*:

“The sample is dipped into the immersion chamber and scanned through the light-sheet plane during the tile acquisition in a stepwise motion using the z-stage (ASI LS-50, 50 mm travel range with optical linear encoder and sub-micron accuracy).”

We hope this answers the question.

8. A paragraph dedicated to limitations and future research is clearly absent from the manuscript. Addressing these limitations would contribute to a more comprehensive presentation.

Reply: Thank you for this suggestion. We have now added a paragraph “Current limitations” to the end of the *Results* section, p. 8, where we summarize our simulations regarding the role of spherical aberrations and objective NA. The new paragraph reads:

“*Current limitations*

*Due to our design choices, we use off-the-shelf air objectives in both excitation and detection arms for compatibility with multiple clearing protocols and quick exchange of imaging media, while keeping the system’s cost low. Thus, the optical resolution of our current design suffers from spherical aberration from the medium index mismatch. Our simulations show that both excitation and detection arms operate in a diffraction-limited regime only with air objectives of NA up to 0.15 and immersion medium thickness up to 15 mm (**SI Notes 3 and 4**). Therefore, increasing the NA of detection objectives above 0.15 achieves higher resolution only for sample (incl. immersion medium) thinner than 15 mm (a “15-15” rule). The simulated PSF size is nearly independent of the refractive index of the medium in the practically relevant range ($n=1.33-1.99$, **SI Table 11**).*

It should be noted that “field flatness” derived from the contrast maps of objectives in our testing method should not be confused with Petzval field curvature. The field flatness profile (2D) is computed from the best-focus surface (in 3D) of the objective under testing conditions. Depending on the chosen contrast criterion, the location of the best-focus surface can be slightly different. The best-focus surface becomes more curved in the presence of planar refractive interface (cuvette wall and immersion medium) due to the presence of coma, astigmatism, and non-telecentric design of some objectives, which together bend the best-focus surface relative to the ideal conditions (air). On the contrary, the well-known Petzval field curvature does not depend on the presence of planar interfaces in the system, and it is independent of spherical aberration, coma, or astigmatism. The best-focus surface we measured in this work is similar to the concept of focal surface in optics.”

In addition, we expanded the *Discussion* on these matters, p.9:

“We thus encourage companies that design and produce microscopy objectives to develop long-working distance air objectives with NA 0.3 to 0.5, magnification 5x to 20x, and build-in compensation of strong spherical aberrations (an equivalent of imaging through a glass block 25-35 mm thick). The ExA-SPIM³⁶ system circumvented this problem by using an industrial telecentric lens with the front beam splitter removed (an equivalent of 35 mm glass block), but this lens type is not infinity corrected, has a large footprint and a low magnification range, which limits its use in the emerging field of light-sheet imaging of cleared samples.”

We thank all three reviewers again for their valuable feedback.

REVIEWERS' COMMENTS

Reviewer #1 (Remarks to the Author):

All the concerns raised by the reviewer have been appropriately addressed.

Reviewer #2 (Remarks to the Author):

Co-reviewed with Reviewer #1

Reviewer #3 (Remarks to the Author):

The authors have comprehensively addressed all comments in the revised version of the paper.